# A convex program for bilinear inversion of sparse vectors

**Alireza Aghasi**
Georgia State Business School
GSU, GA
aaghasi@gsu.edu

**Ali Ahmed**
Dept. of Electrical Engineering
ITU, Lahore
ali.ahmed@itu.edu.pk

**Paul Hand**
Dept. of Mathematics and College of Computer and Information Science
Northeastern University, MA
p.hand@northeastern.edu

**Babhru Joshi**
Dept. of Computational and Applied Mathematics
Rice University, TX
babhru.joshi@rice.edu

## Abstract

We consider the bilinear inverse problem of recovering two vectors, $x \in \mathbb{R}^L$ and $w \in \mathbb{R}^L$, from their entrywise product. We consider the case where $x$ and $w$ have known signs and are sparse with respect to known dictionaries of size $K$ and $N$, respectively. Here, $K$ and $N$ may be larger than, smaller than, or equal to $L$. We introduce $\ell_1$-BranchHull, which is a convex program posed in the natural parameter space and does not require an approximate solution or initialization in order to be stated or solved. We study the case where $x$ and $w$ are $S_1$- and $S_2$-sparse with respect to a random dictionary, with the sparse vectors satisfying an effective sparsity condition, and present a recovery guarantee that depends on the number of measurements as $L \geq \Omega(S_1 + S_2) \log^2(K + N)$. Numerical experiments verify that the scaling constant in the theorem is not too large. One application of this problem is the sweep distortion removal task in dielectric imaging, where one of the signals is a nonnegative reflectivity, and the other signal lives in a known subspace, for example that given by dominant wavelet coefficients. We also introduce a variants of $\ell_1$-BranchHull for the purposes of tolerating noise and outliers, and for the purpose of recovering piecewise constant signals. We provide an ADMM implementation of these variants and show they can extract piecewise constant behavior from real images.

## 1 Introduction

We study the problem of recovering two unknown signals $x$ and $w$ in $\mathbb{R}^L$ from observations $y = \mathcal{A}(w, x)$, where $\mathcal{A}$ is a bilinear operator. Let $B \in \mathbb{R}^{L \times K}$ and $C \in \mathbb{R}^{L \times N}$ such that $w = Bh$ and $x = Cm$ with $\|h\|_0 \leq S_1$ and $\|m\|_0 \leq S_2$. Let the bilinear operator $\mathcal{A} : \mathbb{R}^L \times \mathbb{R}^L \to \mathbb{R}^L$ satisfy

$$y = \mathcal{A}(w, x) = w \odot x, \tag{1}$$

where $\odot$ denotes entrywise product. The bilinear inverse problem (BIP) we consider is to find $w$ and $x$ from $y$, $B$, $C$ and sign $(w)$, up to the inherent scaling ambiguity.

BIPs, in general, have many applications in signal processing and machine learning and include fundamental practical problems like phase retrieval (Fienup [1982], Candès and Li [2012], Candès et al. [2013]), blind deconvolution (Ahmed et al. [2014], Stockham et al. [1975], Kundur and Hatzinakos [1996], Aghasi et al. [2016a]), non-negative matrix factorization (Hoyer [2004], Lee and Seung [2001]), self-calibration (Ling and Strohmer [2015]), blind source separation (D. et al. [2005]), dictionary learning (Tosic and Frossard [2011]), etc. These problems are in general challenging and suffer from identifiability issues that make the solution set non-unique and non-convex. A common identifiability issue, also shared by the BIP in (1), is the scaling ambiguity. In particular, if $(\boldsymbol{w}^\natural, \boldsymbol{x}^\natural)$ solves a BIP, then so does $(c\boldsymbol{w}^\natural, c^{-1}\boldsymbol{x}^\natural)$ for any nonzero $c \in \mathbb{R}$. In this paper, we resolve this scaling ambiguity by finding the point in the solution set closest to the origin with respect to the $\ell_1$ norm.

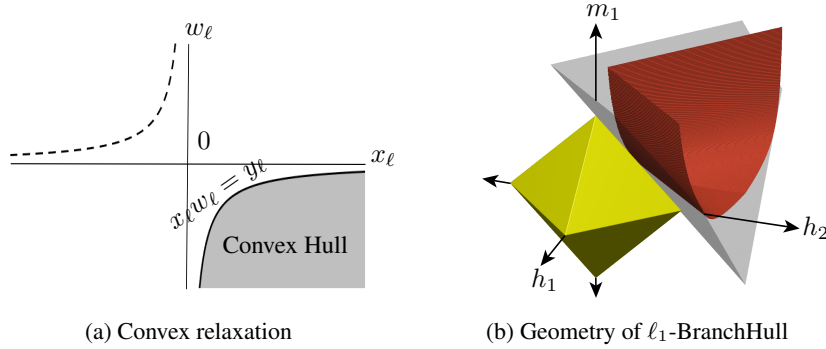

(a) Convex relaxation　　　　　　　(b) Geometry of $\ell_1$-BranchHull

Figure 1: Panel (a) shows the convex hull of the relevant branch of a hyperbola given a measurement $y_\ell$ and the sign information $\mathrm{sign}(w_\ell)$. Panel (b) shows the interaction between the $\ell_1$-ball in the objective of (3) with its feasibility set. The feasibility set is 'pointy' along a hyperbola, which allows for signal recovery where the $\ell_1$ ball touches it. The gray hyperplane segments correspond to linearizations of the hyperbolic measurements, which is an important component of our recovery proof.

Another identifiability issue of the BIP in (1) is if $(\boldsymbol{w}^\natural, \boldsymbol{x}^\natural)$ solves (1), then so does $(\mathbf{1}, \boldsymbol{w}^\natural \odot \boldsymbol{x}^\natural)$, where $\mathbf{1}$ is the vector of ones. In prior works like Ahmed et al. [2014], which studies the blind deconvolution problem and is a BIP in the Fourier Domain, the identifiability issue is resolved by assuming the signals live in a known subspace. In comparison to Ahmed et al. [2014], we resolve the identifiability issue with a much weaker structural assumption of sparsity in known bases at the cost of known signs; justified in actual applications, especially, in imaging. Natural choices for such bases include the standard basis, the Discrete Cosine Transform (DCT) basis, and a wavelet basis.

Recent work on sparse rank-1 matrix recovery problem in Lee et al. [2017], which is motivated by considering the lifted version of the sparse blind deconvolution problem, provides an exact recovery guarantee of the sparse vectors $\boldsymbol{h}$ and $\boldsymbol{m}$ that satisfy a "peakiness" condition, i.e. $\min\{\|\boldsymbol{h}\|_\infty, \|\boldsymbol{m}\|_\infty\} \geq c$ for some absolute constant $c \in \mathbb{R}$. This result holds with high probability for random measurements if the number of measurement, up to a log factor, satisfy $L \geq \Omega(S_1 + S_2)$. For general vectors without the peakiness condition, the same work shows exact recovery is possible if the number of measurements, up to a log factor, satisfy $L \geq \Omega(S_1 S_2)$.

The main contribution of this paper is to introduce an algorithm for the sparse BIP described in (1) which recovers sparse vectors that satisfy a comparable effective sparsity condition. Precisely, we say the sparse vectors $\boldsymbol{h}^\natural$ and $\boldsymbol{m}^\natural$ have comparable effective sparsity if there exist an $\alpha \in \mathbb{R}$ such that

$$\frac{\|\boldsymbol{h}^\natural\|_1}{\|\boldsymbol{h}^\natural\|_2} = \alpha \frac{\|\boldsymbol{m}^\natural\|_1}{\|\boldsymbol{m}^\natural\|_2}. \qquad (2)$$

with $\alpha$ satisfying $\frac{1}{C} \leq \alpha \leq C$ for some $C \in R^+$. Intuitively, the ratios $\frac{\|\boldsymbol{h}^\natural\|_1}{\|\boldsymbol{h}^\natural\|_2}$ and $\frac{\|\boldsymbol{m}^\natural\|_1}{\|\boldsymbol{m}^\natural\|_2}$ are about the same if the sparsity levels of $\boldsymbol{h}^\natural$ and $\boldsymbol{m}^\natural$ are close and the magnitudes of the nonzero entries of $\boldsymbol{h}^\natural$ and $\boldsymbol{m}^\natural$ are about the same. Under this assumption on the sparse signals, we present a convex program stated in the natural parameter space, which in the noiseless setting with random $\boldsymbol{B}$ and

$C$, exactly recovers the sparse vectors with at most $S_1 + S_2$ combined nonzero entries with high probability if the number measurements satisfy $L \geq \Omega(S_1 + S_2) \log^2(K + N)$.

## 1.1 Convex program and main results

We introduce a convex program written in the natural parameter space for the bilinear inverse problem described in (1). Let $(\boldsymbol{h}^\natural, \boldsymbol{m}^\natural) \in \mathbb{R}^K \times \mathbb{R}^N$ with $\|\boldsymbol{h}^\natural\|_0 \leq S_1$ and $\|\boldsymbol{m}^\natural\|_0 \leq S_2$. Let $w_\ell = \boldsymbol{b}_\ell^\mathsf{T} \boldsymbol{h}^\natural$, $x_\ell = \boldsymbol{c}_\ell^\mathsf{T} \boldsymbol{m}^\natural$ and $y_\ell = \boldsymbol{b}_\ell^\mathsf{T} \boldsymbol{h}^\natural \boldsymbol{c}_\ell^\mathsf{T} \boldsymbol{m}^\natural$, where $\boldsymbol{b}_\ell^\mathsf{T}$ and $\boldsymbol{c}_\ell^\mathsf{T}$ are the $\ell$th row of $\boldsymbol{B}$ and $\boldsymbol{C}$. Also, let $\boldsymbol{s} = \mathrm{sign}(\boldsymbol{y})$ and $\boldsymbol{t} = \mathrm{sign}(\boldsymbol{B}\boldsymbol{h}^\natural)$. The convex program we consider to recover $(\boldsymbol{h}^\natural, \boldsymbol{m}^\natural)$ is the $\ell_1$-BranchHull program

$$\ell_1\text{-BH}: \quad \underset{\boldsymbol{h} \in \mathbb{R}^K, \boldsymbol{m} \in \mathbb{R}^N}{\text{minimize}} \|\boldsymbol{h}\|_1 + \|\boldsymbol{m}\|_1 \quad \text{subject to } s_\ell(\boldsymbol{b}_\ell^\mathsf{T} \boldsymbol{h} \boldsymbol{c}_\ell^\mathsf{T} \boldsymbol{m}) \geq |y_\ell| \quad (3)$$

$$t_\ell \boldsymbol{b}_\ell^\mathsf{T} \boldsymbol{h} \geq 0, \quad \ell = 1, 2, \ldots, L.$$

The motivation for the feasible set in program (3) follows from the observation that each measurement $y_\ell = w_\ell \cdot x_\ell$ defines a hyperbola in $\mathbb{R}^2$. As shown in Figure (1a), the sign information $t_\ell = w_\ell$ restricts $(w_\ell, x_\ell)$ to one of the branch of the hyperbola. The feasible set in (3) corresponds to the convex hull of particular branches of the hyperbola for each $y_\ell$. This also implies that the feasible set is convex as it is the intersection of $L$ convex sets.

The objective function in (3) is an $\ell_1$ minimization over $(\boldsymbol{h}, \boldsymbol{m})$ that finds a sparse point $(\hat{\boldsymbol{h}}, \hat{\boldsymbol{m}})$ with $\|\hat{\boldsymbol{h}}\|_1 = \|\hat{\boldsymbol{m}}\|_1$. Geometrically, this happens as the solution lies at the intersection of the $\ell_1$-ball, and the hyperbolic curve (constraint) as shown in Figure 1a and 1b. So, the minimizer of (3), under successful recovery, is $\left( \boldsymbol{h}^\natural \sqrt{\frac{\|\boldsymbol{m}^\natural\|_1}{\|\boldsymbol{h}^\natural\|_1}}, \boldsymbol{m}^\natural \sqrt{\frac{\|\boldsymbol{h}^\natural\|_1}{\|\boldsymbol{m}^\natural\|_1}} \right)$.

Our main result is that under the structural assumptions that $\boldsymbol{w}$ and $\boldsymbol{x}$ live in random subspaces with $(\boldsymbol{h}^\natural, \boldsymbol{m}^\natural)$ containing at most $S_1 + S_2$ non zero entries and $(\boldsymbol{h}^\natural, \boldsymbol{m}^\natural)$ satisfing the effective sparsity condition (2), the $\ell_1$-BranchHull program (3) recovers $\boldsymbol{h}^\natural$, and $\boldsymbol{m}^\natural$ (to within the scaling ambiguity) with high probability, provided the number of measurements, up to log factors, satisfy $L \geq \Omega(S_1 + S_2) \log^2(K + N)$.

**Theorem 1.** *Suppose we observe the pointwise product of two vectors $\boldsymbol{B}\boldsymbol{h}^\natural$, and $\boldsymbol{C}\boldsymbol{m}^\natural$ through a bilinear measurement model in* (1)*, where $\boldsymbol{B}$, and $\boldsymbol{C}$ are standard Gaussian random matrices. If $(\boldsymbol{h}^\natural, \boldsymbol{m}^\natural)$ satisfy* (2)*, then the $\ell_1$-BranchHull program* (3) *uniquely recovers $\left( \boldsymbol{h}^\natural \sqrt{\frac{\|\boldsymbol{m}^\natural\|_1}{\|\boldsymbol{h}^\natural\|_1}}, \boldsymbol{m}^\natural \sqrt{\frac{\|\boldsymbol{h}^\natural\|_1}{\|\boldsymbol{m}^\natural\|_1}} \right)$ whenever $L \geq C \left( \sqrt{S_1 + S_2} \log(K + N) + t \right)^2$ for any $t \geq 0$ with probability at least $1 - \mathrm{e}^{-2Lt^2}$. Here $C$ is an absolute constant.*

## 1.2 Prior art for bilinear inverse problems

Recent approaches to solving bilinear inverse problems like blind deconvolution and phase retrieval have been to lift the problems into a low rank matrix recovery task or to formulate an optimization programs in the natural parameter space. Lifting transforms the problem of recovering $\boldsymbol{h} \in \mathbb{R}^K$ and $\boldsymbol{m} \in \mathbb{R}^N$ from bilinear measurements to the problem of recovering a low rank matrix $\boldsymbol{h}\boldsymbol{m}^\mathsf{T}$ from linear measurements. The low rank matrix can then be recovered using a semidefinite program. The result in Ahmed et al. [2014] for blind deconvolution showed that if $\boldsymbol{h}$ and $\boldsymbol{m}$ are representations of the target signals with respect to Fourier and Gaussian subspaces, respectively, then the lifting method successfully recovers the low rank matrix. The recovery occurs with high probability under near optimal sample complexity. Unfortunately, solving the semidefinite program is prohibitively computationally expensive because they operate in high-dimension space. Also, it is not clear how to enforce additional structure like sparsity of $\boldsymbol{h}$ and $\boldsymbol{m}$ in the lifted formulation in a way that allows optimal sample complexity (Li and Voroninski [2013], Oymak et al. [2015]).

In comparison to the lifting approach for blind deconvolution and phase retrieval, methods that formulate an algorithm in the natural parameter space like alternating minimization and gradient descent based method are computationally efficient and also enjoy rigorous recovery guarantees under optimal or near optimal sample complexity (Li et al. [2016], Candès et al. [2015], Netrapalli et al. [2013], Sun et al. [2016]). In fact, the work in Lee et al. [2017] for sparse blind deconvolution

is based on alternating minimization. In the paper, the authors use an alternating minimization that successively approximate the sparse vectors while enforcing the low rank property of the lifted matrix. However, because these methods are non-convex, convergence to the global optimal requires a good initialization (Tu et al. [2015], Chen and Candes [2015], Li et al. [2016]).

Other approaches that operate in the natural parameter space include PhaseMax (Bahmani and Romberg [2016], Goldstein and Studer [2016]) and BranchHull (Aghasi et al. [2016b]). PhaseMax is a linear program which has been proven to find the target signal in phase retrieval under optimal sample complexity if a good anchor vector is available. As with alternating minimization and gradient descent based approach, PhaseMax requires a good initialization. However, in PhaseMax the initialization is part of the optimization program but in alternating minimization the initialization is part of the algorithmic implementation. BranchHull is a convex program which solves the BIP described in (3) excluding the sparsity assumption under optimal sample complexity. Like the $\ell_1$-BranchHull presented in this paper, BranchHull does not require an initialization but requires the sign information of the signals.

The $\ell_1$-BranchHull program (3) combines strengths of both the lifting method and the gradient descent based method. Specifically, the $\ell_1$-BranchHull program is a convex program that operates in the natural parameter space, without a need for an initialization, and without restrictive assumptions on the class of recoverable signals. These strengths are achieved at the cost of the sign information of the target signals $\boldsymbol{w}$ and $\boldsymbol{x}$. However, the sign assumption can be justified in imaging applications where the goal might be to recover pixel values of a target image, which are non-negative. Also, as in PhaseMax, the sign information can be thought of as an anchor vector which anchors the solution to one of the branches of the $L$ hyperbolic measurements.

### 1.3 Extension to noise and outlier

Extending the theory of the $\ell_1$-BranchHull program (3) to the case with noise is important as most real data contain significant noise. Formulation 3 may be particularly susceptible to noise that changes the sign of even a single measurement. For the bilinear inverse problem as described in (1) with small dense noise and arbitrary outliers, we propose the following robust $\ell_1$-BranchHull program

$$\text{RBH:} \quad \underset{\boldsymbol{h}\in\mathbb{R}^K, \boldsymbol{m}\in\mathbb{R}^N, \boldsymbol{\xi}\in\mathbb{R}^L}{\text{minimize}} \|\boldsymbol{h}\|_1 + \|\boldsymbol{m}\|_1 + \lambda\|\boldsymbol{\xi}\|_1 \quad \text{subject to } s_\ell(\boldsymbol{c}_\ell^\mathsf{T}\boldsymbol{m} + \xi_\ell)\boldsymbol{b}_\ell^\mathsf{T}\boldsymbol{h} \geq |y_\ell|, \quad (4)$$

$$t_\ell \boldsymbol{b}_\ell^\mathsf{T}\boldsymbol{h} \geq 0, \quad \ell = 1, \dots, L.$$

The slack variable $\boldsymbol{\xi}$ controls the shape of the feasible set. For measurements $y_\ell$ with incorrect sign, the corresponding slack variables $\xi_\ell$ shifts the feasible set so that the target signal is feasible. In the outlier case, the $\ell_1$ penalty promotes sparsity of slack variable $\boldsymbol{\xi}$. We implement a slight variation of the above program, detailed in Section 1.4, to remove distortions from real and synthetic images.

### 1.4 Total variation extension of $\ell_1$-BranchHull

The robust $\ell_1$-BranchHull program (4) is flexible and can be altered to remove distortions from an otherwise piecewise constant signal. In the case where $\boldsymbol{w} = \boldsymbol{B}\boldsymbol{h}^\natural$ is a piecewise constant signal, $\boldsymbol{x} = \boldsymbol{C}\boldsymbol{m}^\natural$ is a distortion signal and $\boldsymbol{y} = \boldsymbol{w} \odot \boldsymbol{x}$ is the distorted signal, the total variation version (5) of the robust BranchHull program (4), under successful recovery, produces the piecewise constant signal $\boldsymbol{B}\boldsymbol{h}^\natural$, up to a scaling.

$$\text{TV BH :} \quad \underset{\boldsymbol{h}\in\mathbb{R}^K, \boldsymbol{m}\in\mathbb{R}^N, \boldsymbol{\xi}\in\mathbb{R}^L}{\text{minimize}} \text{TV}(\boldsymbol{B}\boldsymbol{h}) + \|\boldsymbol{m}\|_1 + \lambda\|\boldsymbol{\xi}\|_1 \quad \text{subject to } s_\ell(\xi_\ell + \boldsymbol{c}_\ell^\top\boldsymbol{m})\boldsymbol{b}_\ell^\top\boldsymbol{h} \geq |y_\ell| \quad (5)$$

$$t_\ell \boldsymbol{b}_\ell^\top\boldsymbol{h} \geq 0, \quad \ell = 1, 2, \dots, L.$$

In (5), $\text{TV}(\cdot)$ is a total variation operator and is the $\ell_1$ norm of the vector containing pairwise difference of neighboring elements of the target signal $\boldsymbol{B}\boldsymbol{h}$. We implement (5) to remove distortions from images in Section 3.2.

### 1.5 Notation

Vectors and matrices are written with boldface, while scalars and entries of vectors are written in plain font. For example, $c_\ell$ is the $\ell$the entry of the vector $\boldsymbol{c}$. We write $\mathbf{1}$ as the vector of all ones with

dimensionality appropriate for the context. We write $\boldsymbol{I}_N$ as the $N \times N$ identity matrix. For any $x \in \mathbb{R}$, let $(x)_- \in \mathbb{Z}$ such that $x - 1 < (x)_- \leq x$. For any matrix $\boldsymbol{A}$, let $\|\boldsymbol{A}\|_F$ be the Frobenius norm of $\boldsymbol{A}$. For any vector $\boldsymbol{x}$, let $\|\boldsymbol{x}\|_0$ be the number of non-zero entries in $\boldsymbol{x}$. For $\boldsymbol{x} \in \mathbb{R}^K$ and $\boldsymbol{y} \in \mathbb{R}^N$, $(\boldsymbol{x}, \boldsymbol{y})$ is the corresponding vector in $\mathbb{R}^K \times \mathbb{R}^N$, and $\langle (\boldsymbol{x}_1, \boldsymbol{y}_1), (\boldsymbol{x}_2, \boldsymbol{y}_2) \rangle = \langle \boldsymbol{x}_1, \boldsymbol{x}_2 \rangle + \langle \boldsymbol{y}_1, \boldsymbol{y}_2 \rangle$. For a set $\mathcal{A} \subset \mathbb{R}^m$, and a vector $\boldsymbol{a} \in \mathbb{R}^m$, we define by $\boldsymbol{a} \oplus \mathcal{A}$, a set obtained by incrementing every element of $\mathcal{A}$ by $\boldsymbol{a}$.

## 2   Algorithm

In this section, we present an Alternating Direction Method of Multipliers (ADMM) implementation of an extension of the robust $\ell_1$-BranchHull program (4). The ADMM implementation of the $\ell_1$-BranchHull program (3) is similar to the ADMM implementation of (6) and we leave it to the readers. The extension of the robust $\ell_1$-BranchHull program we consider is

$$\underset{\boldsymbol{h} \in \mathbb{R}^K, \boldsymbol{m} \in \mathbb{R}^N, \boldsymbol{\xi} \in \mathbb{R}^L}{\text{minimize}} \ \|\boldsymbol{P}\boldsymbol{h}\|_1 + \|\boldsymbol{m}\|_1 + \lambda\|\boldsymbol{\xi}\|_1 \quad \text{subject to} \ \ s_\ell(\xi_\ell + \boldsymbol{c}_\ell^\top \boldsymbol{m})\boldsymbol{b}_\ell^\top \boldsymbol{h} \geq |y_\ell| \tag{6}$$

$$t_\ell \boldsymbol{b}_\ell^\top \boldsymbol{h} \geq 0, \quad \ell = 1, 2, \ldots, L,$$

where $\boldsymbol{P} \in \mathbb{R}^{J \times K}$ for some $J \in \mathbb{Z}$. The above extension reduces to the robust $\ell_1$-BranchHull program if $\boldsymbol{P} = \boldsymbol{I}_K$. Recalling that $\boldsymbol{w} = \boldsymbol{B}\boldsymbol{h}$ and $\boldsymbol{x} = \boldsymbol{C}\boldsymbol{m}$, we make use of the following notations

$$\boldsymbol{u} = \begin{pmatrix} \boldsymbol{x} \\ \boldsymbol{w} \\ \boldsymbol{\xi} \end{pmatrix}, \quad \boldsymbol{v} = \begin{pmatrix} \boldsymbol{m} \\ \boldsymbol{h} \\ \lambda\boldsymbol{\xi} \end{pmatrix}, \quad \boldsymbol{E} = \begin{pmatrix} \boldsymbol{C} & 0 & 0 \\ 0 & \boldsymbol{B} & 0 \\ 0 & 0 & \lambda^{-1}\boldsymbol{I}_L \end{pmatrix} \text{ and } \boldsymbol{Q} = \begin{pmatrix} \boldsymbol{I}_N & 0 & 0 \\ 0 & \boldsymbol{P} & 0 \\ 0 & 0 & \boldsymbol{I}_L \end{pmatrix}.$$

Using this notation, our convex program can be compactly written as

$$\underset{\boldsymbol{v} \in \mathbb{R}^{N+K+L}, \boldsymbol{u} \in \mathbb{R}^{3L}}{\text{minimize}} \ \|\boldsymbol{Q}\boldsymbol{v}\|_1 \ \text{subject to} \ \boldsymbol{u} = \boldsymbol{E}\boldsymbol{v}, \ \boldsymbol{u} \in \mathcal{C}.$$

Here $\mathcal{C} = \left\{ (\boldsymbol{x}, \boldsymbol{w}, \boldsymbol{\xi}) \in \mathbb{R}^{3L} \mid s_\ell(\xi_\ell + x_\ell)w_\ell \geq |y_\ell|, \ t_\ell w_\ell \geq 0, \ \ell = 1, \ldots, L \right\}$ is the convex feasible set of (6). Introducing a new variable $\boldsymbol{z}$ the resulting convex program can be written as

$$\underset{\boldsymbol{v}, \boldsymbol{u}, \boldsymbol{z}}{\text{minimize}} \ \|\boldsymbol{z}\|_1 \ \text{subject to} \ \boldsymbol{u} = \boldsymbol{E}\boldsymbol{v}, \ \boldsymbol{Q}\boldsymbol{v} = \boldsymbol{z}, \ \boldsymbol{u} \in \mathcal{C}.$$

We may now form the scaled ADMM steps as follows

$$\boldsymbol{u}_{k+1} = \arg\min_{\boldsymbol{u}} \ \mathcal{I}_\mathcal{C}(\boldsymbol{u}) + \frac{\rho}{2}\|\boldsymbol{u} + \boldsymbol{\alpha}_k - \boldsymbol{E}\boldsymbol{v}_k\|^2 \tag{7}$$

$$\boldsymbol{z}_{k+1} = \arg\min_{\boldsymbol{z}} \ \|\boldsymbol{z}\|_1 + \frac{\rho}{2}\|\boldsymbol{z} + \boldsymbol{\beta}_k - \boldsymbol{Q}\boldsymbol{v}_k\|^2 \tag{8}$$

$$\boldsymbol{v}_{k+1} = \arg\min_{\boldsymbol{v}} \ \frac{\rho}{2}\|\boldsymbol{\alpha}_k + \boldsymbol{u}_{k+1} - \boldsymbol{E}\boldsymbol{v}\|^2 + \frac{\rho}{2}\|\boldsymbol{\beta}_k + \boldsymbol{z}_{k+1} - \boldsymbol{Q}\boldsymbol{v}\|^2, \tag{9}$$

$$\boldsymbol{\alpha}_{k+1} = \boldsymbol{\alpha}_k + \boldsymbol{u}_{k+1} - \boldsymbol{E}\boldsymbol{v}_{k+1},$$

$$\boldsymbol{\beta}_{k+1} = \boldsymbol{\beta}_k + \boldsymbol{v}_{k+1} - \boldsymbol{Q}\boldsymbol{v}_{k+1}.$$

where $\mathcal{I}_\mathcal{C}(\cdot)$ in (7) is the indicator function on $\mathcal{C}$ such that $\mathcal{I}_\mathcal{C}(u) = 0$ if $u \in \mathcal{C}$ and infinity otherwise. We would like to note that the first three steps of the proposed ADMM scheme can be presented in closed form. The update in (7) is the following projection

$$\boldsymbol{u}_{k+1} = \text{proj}_\mathcal{C}(\boldsymbol{E}\boldsymbol{v}_k - \boldsymbol{\alpha}_k),$$

where $\text{proj}_\mathcal{C}(\boldsymbol{v})$ is the projection of $\boldsymbol{v}$ onto $\mathcal{C}$. Details of computing the projection onto $\mathcal{C}$ are presented in the Supplementary material. The update in (8) can be written in terms of the soft-thresholding operator

$$\boldsymbol{z}_{k+1} = S_{1/\rho}(\boldsymbol{Q}\boldsymbol{v}_k - \boldsymbol{\beta}_k), \qquad \text{where} \quad (S_c(\boldsymbol{z}))_i = \begin{cases} z_i - c & z_i > c \\ 0 & |z_i| \leq c \\ z_i + c & z_i < -c \end{cases},$$

where $c > 0$ and $(S_c(\boldsymbol{z}))_i$ is the $i$th entry of $S_c(\boldsymbol{z})$. Finally, the update in (9) takes the following form

$$\boldsymbol{v}_{k+1} = \left(\boldsymbol{E}^\top\boldsymbol{E} + \boldsymbol{Q}^\top\boldsymbol{Q}\right)^{-1}\left(\boldsymbol{E}^\top(\boldsymbol{\alpha}_k + \boldsymbol{u}_{k+1}) + \boldsymbol{Q}^\top(\boldsymbol{\beta}_k + \boldsymbol{z}_{k+1})\right).$$

In our implementation of the ADMM scheme, we initialize the algorithm with the $\boldsymbol{v}_0 = \boldsymbol{0}$, $\boldsymbol{\alpha}_0 = \boldsymbol{0}$, $\boldsymbol{\beta}_0 = \boldsymbol{0}$.

## 3 Numerical Experiments

In this section, we provide numerical experiments on synthetic and real data where the signals follow the multiplicative model (1), which is compatible with physics of lighting (Hold [1986]). This is in contrast to well-known methods for image de-illumination like He et al. [2011] where the external light has an additive contribution to the image. Other methods like Chen et al. [2006] work with additive models by working with the images in the log domain, while we directly work with the multiplicative model in a robust-to-noise way. The experiment on real data presented in this section shows total variation $\ell_1$-BranchHull program can be used to remove distortions from an image. The synthetic experiment numerically verifies Theorem 1 with a low scaling constant.

### 3.1 Phase Portrait

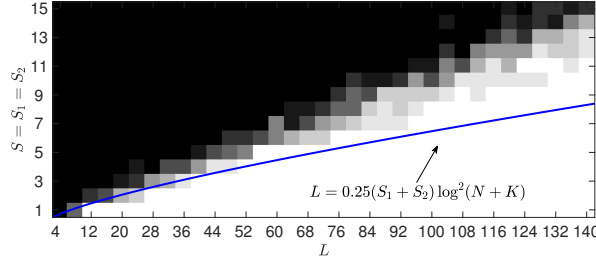

Figure 2: The empirical recovery probability from synthetic data with sparsity level $S$ as a function of total number of measurements $L$. Each block correspond to the average from 10 independent trials. White blocks correspond to successful recovery and black blocks correspond to unsuccessful recovery. The area to the right of the line satisfies $L > 0.25(S_1 + S_2)\log^2(N + K)$.

We first show a phase portrait that verifies Theorem 1. Consider the following measurements: fix $N \in \{20, 40, \ldots, 300\}$, $L \in \{4, 8, \ldots, 140\}$ and let $K = N$. Let the target signal $(\boldsymbol{h}^\natural, \boldsymbol{m}^\natural) \in \mathbb{R}^K \times \mathbb{R}^N$ be such that both $\boldsymbol{h}^\natural$ and $\boldsymbol{m}^\natural$ have $0.05N$ non-zero entries with the nonzero indices randomly selected and set to $\pm 1$. Let $S_1$ and $S_2$ be the number of nonzero entries in $\boldsymbol{h}^\natural$ and $\boldsymbol{m}^\natural$, respectively. Let $\boldsymbol{B} \in \mathbb{R}^{L \times K}$ and $\boldsymbol{C} \in \mathbb{R}^{L \times N}$ such that $B_{ij} \sim \frac{1}{\sqrt{L}}\mathcal{N}(0,1)$ and $C_{ij} \sim \frac{1}{\sqrt{L}}\mathcal{N}(0,1)$. Lastly, let $\boldsymbol{y} = \boldsymbol{B}\boldsymbol{h}^\natural \odot \boldsymbol{C}\boldsymbol{m}^\natural$ and $\boldsymbol{t} = \text{sign}(\boldsymbol{B}\boldsymbol{h}^\natural)$.

Figure 2 shows the fraction of successful recoveries from 10 independent trials using (3) for the bilinear inverse problem (1) from data as described above. Let $(\hat{\boldsymbol{h}}, \hat{\boldsymbol{m}})$ be the output of (3) and let $(\tilde{\boldsymbol{h}}, \tilde{\boldsymbol{m}})$ be the candidate minimizer. We solve (3) using an ADMM implementation similar to the ADMM implementation detailed in Section 2 with the step size parameter $\rho = 1$. For each trial, we say (3) successfully recovers the target signal if $\|(\hat{\boldsymbol{h}}, \hat{\boldsymbol{m}}) - (\tilde{\boldsymbol{h}}, \tilde{\boldsymbol{m}})\|_2 < 10^{-10}$. Black squares correspond to no successful recovery and white squares correspond to 100% successful recovery. The line corresponds to $L = C(S_1 + S_2)\log^2(K + N)$ with $C = 0.25$ and indicates that the sample complexity constant in Theorem 1 is not very large.

### 3.2 Distortion removal from images

We use the total variation BranchHull program (5) to remove distortions from real images $\tilde{\boldsymbol{y}} \in \mathbb{R}^{p \times q}$. In the experiments, The observation $\boldsymbol{y} \in \mathbb{R}^L$ is the column-wise vectorization of the image $\tilde{\boldsymbol{y}}$, the target signal $\boldsymbol{w} = \boldsymbol{B}\boldsymbol{h}$ is the vectorization of the piecewise constant image and $\boldsymbol{x} = \boldsymbol{C}\boldsymbol{m}$ corresponds to the distortions in the image. We use (5) to recover piecewise constant target images like in the foreground of Figure 3a with $\text{TV}(\boldsymbol{B}\boldsymbol{h}) = \|\boldsymbol{D}\boldsymbol{B}\boldsymbol{h}\|_1$, where $\boldsymbol{D} = \begin{bmatrix} \boldsymbol{D}_v \\ \boldsymbol{D}_h \end{bmatrix}$ in block form. Here, $\boldsymbol{D}_v \in \mathbb{R}^{(L-q) \times L}$ and $\boldsymbol{D}_h \in \mathbb{R}^{(L-p) \times L}$ with

$$(\boldsymbol{D}_v)_{ij} = \begin{cases} -1 & \text{if } j = i + \left(\frac{i-1}{p-1}\right)_- \\ 1 & \text{if } j = i + 1 + \left(\frac{i-1}{p-1}\right)_- \\ 0 & \text{otherwise} \end{cases}, \quad (\boldsymbol{D}_h)_{ij} = \begin{cases} -1 & \text{if } j = i \\ 1 & \text{if } j = i + p \\ 0 & \text{otherwise} \end{cases}.$$

Lastly, we solve (5) using the ADMM algorithm detailed in Section 2 with $\boldsymbol{P} = \boldsymbol{DB}$.

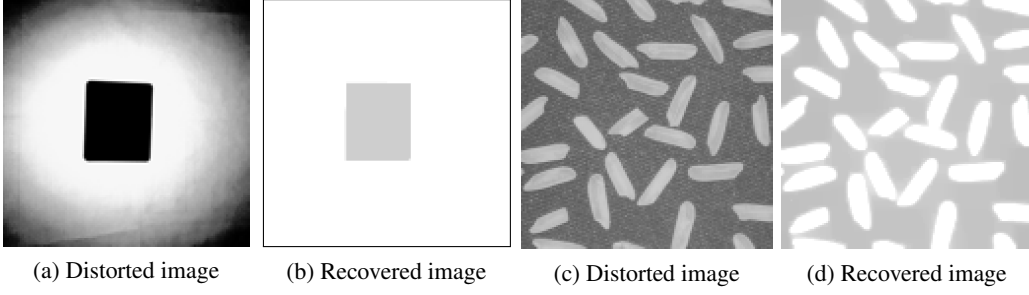

| (a) Distorted image | (b) Recovered image | (c) Distorted image | (d) Recovered image |

Figure 3: Panel (a) shows an image of a mousepad with distortions and panel(b) is the piecewise constant image recovered using total variation $\ell_1$-BranchHull. Similarly, panel (d) shows an image containing rice grains and panel (e) is the recovered image.

We now show two experiments on real images. The first image, shown in Figure 3a, was captured using a camera and resized to a $115 \times 115$ image. The measurement $\boldsymbol{y} \in \mathbb{R}^L$ is the vectorization of the image with $L = 13225$. Let $\boldsymbol{B}$ be the $L \times L$ identity matrix. Let $\boldsymbol{F}$ be the $L \times L$ inverse DCT matrix. Let $\boldsymbol{C} \in \mathbb{R}^{L \times 300}$ with the first column set to $\mathbf{1}$ and remaining columns randomly selected from columns of $\boldsymbol{F}$ without replacement. The matrix $\boldsymbol{C}$ is scaled so that $\|\boldsymbol{C}\|_F = \|\boldsymbol{B}\|_F = \sqrt{L}$. The vector of known sign $\boldsymbol{t}$ is set to $\mathbf{1}$. Let $(\hat{\boldsymbol{h}}, \hat{\boldsymbol{m}}, \hat{\boldsymbol{\xi}})$ be the output of (5) with $\lambda = 10^3$ and $\rho = 10^{-4}$. Figure 3b corresponds to $\boldsymbol{B}\hat{\boldsymbol{h}}$ and shows that the object in the center was successfully recovered.

The second real image, shown in Figure 3c, is an image of rice grains. The size of the image is $128 \times 128$. The measurement $\boldsymbol{y} \in \mathbb{R}^L$ is the vectorization of the image with $L = 16384$. Let $\boldsymbol{B}$ be the $L \times L$ identity matrix. Let $\boldsymbol{C} \in \mathbb{R}^{L \times 50}$ with the first column set to $\mathbf{1}$. The remaining columns of $\boldsymbol{C}$ are sampled from Bessel function of the first kind $J_\nu(\gamma)$ with each column corresponding to a fixed $\gamma \in \mathbb{R}$. Specifically, fix $\boldsymbol{g} \in \mathbb{R}^L$ with $g_i = -9 + 14\frac{i-1}{L-1}$. For each remaining column $\boldsymbol{c}$ of $\boldsymbol{C}$, fix $\boldsymbol{\zeta} \sim \mathcal{N}(\mathbf{0}, \mathrm{I}_3)$ and let $c_i = J_{\frac{g_i}{6+0.1|\zeta_1|}+5|\zeta_2|}(0.1 + 10|\zeta_3|)$. The matrix $\boldsymbol{C}$ is scaled so that $\|\boldsymbol{C}\|_F = \|\boldsymbol{B}\|_F = \sqrt{L}$. The vector of known sign $\boldsymbol{t}$ is set to $\mathbf{1}$. Let $(\hat{\boldsymbol{h}}, \hat{\boldsymbol{m}}, \hat{\boldsymbol{\xi}})$ be the output of (5) with $\lambda = 10^3$ and $\rho = 10^{-7}$. Figure 3d corresponds to $\boldsymbol{B}\hat{\boldsymbol{h}}$.

## 4  Proof Outline

In this section, we provide a proof of Theorem 1 by considering a related linear program with larger feasible set. Let $(\boldsymbol{h}^\natural, \boldsymbol{m}^\natural) \in \mathbb{R}^K \times \mathbb{R}^N$ with $\|\boldsymbol{h}^\natural\|_0 \leq S_1$ and $\|\boldsymbol{m}^\natural\|_0 \leq S_2$. Let $w_\ell = \boldsymbol{b}_\ell^\mathsf{T} \boldsymbol{h}^\natural$, $x_\ell = \boldsymbol{c}_\ell^\mathsf{T} \boldsymbol{m}^\natural$ and $y_\ell = \boldsymbol{b}_\ell^\mathsf{T} \boldsymbol{h}^\natural \cdot \boldsymbol{c}_\ell^\mathsf{T} \boldsymbol{m}^\natural$. Also, let $\boldsymbol{s} = \operatorname{sign}(\boldsymbol{y})$ and $\boldsymbol{t} = \operatorname{sign}(\boldsymbol{B}\boldsymbol{h}^\natural)$. We will shows that the (3) recovers $(\tilde{\boldsymbol{h}}, \tilde{\boldsymbol{m}})$ such that $(\tilde{\boldsymbol{h}}, \tilde{\boldsymbol{m}}) = \left( \boldsymbol{h}^\natural \sqrt{\frac{\|\boldsymbol{m}^\natural\|_1}{\|\boldsymbol{h}^\natural\|_1}}, \boldsymbol{m}^\natural \sqrt{\frac{\|\boldsymbol{h}^\natural\|_1}{\|\boldsymbol{m}^\natural\|_1}} \right)$.

Consider program (10) which has a linear constraint set that contains the feasible set of the $\ell_1$-BrachHull program (3).

$$\text{LP}: \quad \underset{\boldsymbol{h} \in \mathbb{R}^K, \boldsymbol{m} \in \mathbb{R}^N}{\text{minimize}} \|\boldsymbol{h}\|_1 + \|\boldsymbol{m}\|_1 \text{subject to} \ s_\ell(\boldsymbol{b}_\ell^\mathsf{T} \boldsymbol{h} \boldsymbol{c}_\ell^\mathsf{T} \tilde{\boldsymbol{m}} + \boldsymbol{b}_\ell^\mathsf{T} \tilde{\boldsymbol{h}} \boldsymbol{c}_\ell^\mathsf{T} \boldsymbol{m}) \geq 2|y_\ell| \quad (10)$$

$$\ell = 1, 2, \ldots, L,$$

Let

$$\mathcal{S} := \left\{ (\boldsymbol{h}, \boldsymbol{m}) \in \mathbb{R}^K \times \mathbb{R}^N \mid (\boldsymbol{h}, \boldsymbol{m}) = \alpha(-\tilde{\boldsymbol{h}}, \tilde{\boldsymbol{m}}), \text{ and } \alpha \in [-1, 1] \right\}. \quad (11)$$

Observe that if $(\tilde{\boldsymbol{h}}, \tilde{\boldsymbol{m}})$ is a minimizer of (10) then so are all the points in the set $(\tilde{\boldsymbol{h}}, \tilde{\boldsymbol{m}}) \oplus \mathcal{S}$.

**Lemma 1.** *If the optimization program* (10) *recovers* $(\boldsymbol{h}, \boldsymbol{m}) \in (\tilde{\boldsymbol{h}}, \tilde{\boldsymbol{m}}) \oplus \mathcal{S}$, *then the BranchHull program* (3) *recovers* $(\tilde{\boldsymbol{h}}, \tilde{\boldsymbol{m}})$.

A proof of Lemma 1, provided in Supplementary material, follows from the observations that the feasible set of (10) contains the feasible set of (3) and $(\tilde{h}, \tilde{m})$ is the only feasible point in (3) among all $(h, m) \in (\tilde{h}, \tilde{m}) \oplus \mathcal{S}$.

We now show that the solution of (10) lies in the set $(\tilde{h}, \tilde{m}) \oplus \mathcal{S}$. Let $a_\ell^\mathsf{T} = (c_\ell^\mathsf{T} \tilde{m} b_\ell^\mathsf{T}, b_\ell^\mathsf{T} \tilde{h} c_\ell^\mathsf{T}) \in \mathbb{R}^{K+N}$ denote the $\ell$th row of a matrix $A$. The linear constraint in (10) are now simply $s \odot A(h, m) \geq 2|y|$. Note that $\mathcal{S} \subset \mathcal{N} := \operatorname{span}(-\tilde{h}, \tilde{m}) \subseteq \operatorname{Null}(A)$.

Our strategy will be to show that for any feasible perturbation $(\delta h, \delta m) \in \mathcal{N}_\perp$ the objective of the linear program (10) strictly increases, where $\mathcal{N}_\perp$ is the orthogonal complement of the subspace $\mathcal{N}$. This will be equivalent to showing that the solution of (10) lies in the set $(\tilde{h}, \tilde{m}) \oplus \mathcal{S}$.

The subgradient of the $\ell_1$-norm at the proposed solution $(\tilde{h}, \tilde{m})$ is

$$\partial \|(\tilde{h}, \tilde{m})\|_1 := \{g \in \mathbb{R}^{K+N} : \|g\|_\infty \leq 1 \text{ and } g_{\Gamma_h} = \operatorname{sign}(h_{\Gamma_h}^\natural), \, g_{\Gamma_m} = \operatorname{sign}(m_{\Gamma_m}^\natural)\},$$

where $\Gamma_h$, and $\Gamma_m$ denote the support of non-zeros in $h^\natural$, and $m^\natural$, respectively. To show the linear program converges to a solution $(\hat{h}, \hat{m}) \in (\tilde{h}, \tilde{m}) \oplus \mathcal{S}$, it suffices to show that the set of following descent directions

$$\begin{aligned}
&\left\{(\delta h, \delta m) \in \mathcal{N}_\perp : \langle g, (\delta h, \delta m)\rangle \leq 0, \, \forall g \in \partial \|(\tilde{h}, \tilde{m})\|_1\right\} \\
&\subseteq \left\{(\delta h, \delta m) \in \mathcal{N}_\perp : \langle g_{\Gamma_h}, \delta h_{\Gamma_h}\rangle + \langle g_{\Gamma_m}, \delta m_{\Gamma_m}\rangle + \|(\delta h_{\Gamma_h^c}, \delta m_{\Gamma_m^c})\|_1 \leq 0\right\} \\
&\subseteq \left\{(\delta h, \delta m) \in \mathcal{N}_\perp : -\|g_{\Gamma_h \cup \Gamma_m}\|_2 \|(\delta h_{\Gamma_h}, \delta m_{\Gamma_m})\|_2 + \|(\delta h_{\Gamma_h^c}, \delta m_{\Gamma_m^c})\|_1 \leq 0\right\} \\
&= \left\{(\delta h, \delta m) \in \mathcal{N}_\perp : \|(\delta h_{\Gamma_h^c}, \delta m_{\Gamma_m^c})\|_1 \leq \sqrt{S_1 + S_2} \|(\delta h_{\Gamma_h}, \delta m_{\Gamma_m})\|_2\right\} =: \mathcal{D} \qquad (12)
\end{aligned}$$

does not contain any vector $(\delta h, \delta m)$ that is consistent with the constraints. We do this by quantifying the "width" of the set $\mathcal{D}$ through a Rademacher complexity, and a probability that the gradients of the constraint functions lie in a certain half space. This allows us to use small ball method developed in Koltchinskii and Mendelson [2015], Mendelson [2014] to ultimately show that it is highly unlikely to have descent directions in $\mathcal{D}$ that meet the constraints in (10). We now concretely state the definitions of the Rademacher complexity, and probability term mentioned above.

Define linear functions

$$f_\ell(h, m) := \left\langle (b_\ell^\mathsf{T} \tilde{h} c_\ell, c_\ell^\mathsf{T} \tilde{m} b_\ell), (h, m)\right\rangle, \ell = 1, 2, 3, \ldots, L.$$

The linear constraints in the LP (10) are defined these linear functions as $s_\ell f_\ell(h, m) \geq 2|y_\ell|$. The gradients of $f_\ell$ w.r.t. $(h, m)$ at $(\tilde{h}, \tilde{m})$ are then simply $\nabla f_\ell = (\frac{\partial f_\ell(\tilde{h}, \tilde{m})}{\partial h}, \frac{\partial f_\ell(\tilde{h}, \tilde{m})}{\partial m}) = (s_\ell c_\ell^\mathsf{T} \tilde{m} b_\ell, s_\ell b_\ell^\mathsf{T} \tilde{h} c_\ell)$. Define the Rademacher complexity of a set $\mathcal{D} \subset \mathbb{R}^M$ as

$$\mathfrak{C}(\mathcal{D}) := \mathrm{E} \sup_{(h, m) \in \mathcal{D}} \frac{1}{\sqrt{L}} \sum_{\ell=1}^{L} \varepsilon_\ell \left\langle \nabla f_\ell, \frac{(h, m)}{\|(h, m)\|_2}\right\rangle, \qquad (13)$$

where $\varepsilon_1, \varepsilon_2, \ldots, \varepsilon_L$ are iid Rademacher random variables that are independent of everything else. For a set $\mathcal{D}$, the quantity $\mathfrak{C}(\mathcal{D})$ is a measure of width of $\mathcal{D}$ around the origin in terms of the gradients of the constraint functions. For example, an equally distributed random set of gradient functions might lead to a smaller value of $\mathcal{C}(\mathcal{D})$.

Our results also depend on a probability $p_\tau(\mathcal{D})$, and a positive parameter $\tau$ introduced below

$$\mathfrak{p}_\tau(\mathcal{D}) = \inf_{(h, m) \in \mathcal{D}} \mathbb{P}\left(\left\langle \nabla f_\ell, \frac{(h, m)}{\|(h, m)\|_2}\right\rangle \geq \tau\right). \qquad (14)$$

Intuitively, $p_\tau(\mathcal{D})$ quantifies the size of $\mathcal{D}$ through the gradient vectors. For a small enough fixed parameter, a small value of $p_\tau(\mathcal{D})$ means that the $\mathcal{D}$ is mainly invisible to to the gradient vectors.

**Lemma 2.** *Let $\mathcal{D}$ be the set of descent directions, already characterized in* (12)*, for which $\mathfrak{C}(\mathcal{D})$, and $\mathfrak{p}_\tau(\mathcal{D})$ can be determined using* (13)*, and* (14)*. Choose $L \geq \left(\frac{2\mathfrak{C}(\mathcal{D}) + t\tau}{\tau \mathfrak{p}_\tau(\mathcal{D})}\right)^2$ for any $t > 0$. Then the solution $(\hat{h}, \hat{m})$ of the LP in* (10) *lies in the set $(\tilde{h}, \tilde{m}) \oplus \mathcal{S}$ with probability at least $1 - \mathrm{e}^{-2Lt^2}$.*

Proof of this lemma is based on small ball method developed in Koltchinskii and Mendelson [2015], Mendelson [2014] and further studied in Lecué et al. [2018], Lecué and Mendelson [2017]. The proof is mainly repeated using the argument in Bahmani and Romberg [2017], and is provided in the supplementary material for completeness. We now state the main theorem for linear program (10). The theorems states that if the sparse signals satisfy the effective sparsity condition (2) and $L \geq C_t(S_1 + S_2) \log^2(K + N)$, then the minimizer of the linear program (10) is in the set $(\tilde{h}, \tilde{m}) \oplus \mathcal{S}$ with high probability.

**Theorem 2** (Exact recovery). *Suppose we observe pointwise product of two vectors $Bh^\natural$, and $Cm^\natural$ through a bilinear measurement model in* (1)*, where $B$, and $C$ are standard Gaussian random matrices. If $(h^\natural, m^\natural)$ satisfy* (2)*, then the linear program* (10) *recovers $(\hat{h}, \hat{m}) \in (\tilde{h}, \tilde{m}) \oplus \mathcal{S}$ with probability at least $1 - \mathrm{e}^{-2Lt^2}$ whenever $L \geq C\left(\sqrt{S_1 + S_2} \log(K + N) + t\right)^2$, where $C$ is an absolute constant.*

In light of Lemma 2, the proof of Theorem 2 reduces to computing the Rademacher complexity $\mathfrak{C}(\mathcal{D})$ defined in (13), and the tail probability estimate $\mathfrak{p}_\tau(\mathcal{D})$ defined in (14) of the set of descent directions $\mathcal{D}$ defined in (12). The Rademacher complexity is bounded from above by

$$\mathfrak{C}(\mathcal{D}) \leq C \sqrt{\left(\|\tilde{m}\|_2^2 + \|\tilde{h}\|_2^2\right)(S_1 + S_2) \log^2(K + N)}.$$

and for $\tau = \min\{\|\tilde{h}\|_2, \|\tilde{m}\|_2\}$, the tail probability is bounded by $\mathfrak{p}_\tau(\mathcal{D}) \geq \frac{1}{8c^4}$, where both $C$ and $c$ are constants. These bounds are shown in the Supplementary material. The proof of Theorem 1 follows by applying Lemma 1 to Theorem 2.

### Acknowledgements

Ali Ahmed would like to acknowledge the partial support through the grant for the National center of cyber security (NCCS) from HEC, Pakistan. Paul Hand would like to acknowledge funding by the grant NSF DMS-1464525.

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
