[Supplementary Material]

# A   Supplementary material

## A.1   Proof of Lemma 1:

We first show that the feasible set of (3) is contained in the feasible set of (10). We do this by using the fact that a convex set with a smooth boundary is contained in the halfspace defined by the tangent hyperplane at any point of the boundary of the convex set. Consider a point $(\tilde{w}_\ell, \tilde{x}_\ell)$ on the boundary of the convex set defined by the constraints in (3) and observe that

$$\left\{ (w_\ell, x_\ell) \in \mathbb{R}^2 \middle| \begin{matrix} s_\ell w_\ell x_\ell \geq |y_\ell| \\ \mathrm{sign}(w_\ell) = t_\ell \end{matrix} \right\} \subseteq \left\{ (w_\ell, x_\ell) \in \mathbb{R}^2 \middle| \begin{pmatrix} s_\ell \tilde{x}_\ell \\ s_\ell \tilde{w}_\ell \end{pmatrix} \cdot \begin{pmatrix} w_\ell - \tilde{w}_\ell \\ x_\ell - \tilde{x}_\ell \end{pmatrix} \geq 0 \right\}. \tag{15}$$

Plugging in $w_\ell = \boldsymbol{b}_\ell^\mathsf{T} \boldsymbol{h}$ and $x_\ell = \boldsymbol{c}_\ell^\mathsf{T} \boldsymbol{m}$, we have that any feasible $(\boldsymbol{h}, \boldsymbol{m})$ satisfies

$$s_\ell \boldsymbol{c}_\ell^\mathsf{T} \tilde{\boldsymbol{m}} \boldsymbol{b}_\ell^\mathsf{T} \boldsymbol{h} + s_\ell \boldsymbol{b}_\ell^\mathsf{T} \tilde{\boldsymbol{h}} \boldsymbol{c}_\ell^\mathsf{T} \boldsymbol{m} \geq 2|y_\ell|, \quad \ell = 1, \dots, L,$$

which implies $s_\ell (\boldsymbol{b}_\ell^\mathsf{T} \boldsymbol{h} \boldsymbol{c}_\ell^\mathsf{T} \tilde{\boldsymbol{m}} + \boldsymbol{b}_\ell^\mathsf{T} \tilde{\boldsymbol{h}} \boldsymbol{c}_\ell^\mathsf{T} \boldsymbol{m}) \geq 2|y_\ell|$ for all $\ell$. So, the feasible set of (10) contains the feasible set of (3). Lastly, note that among all points $(\boldsymbol{h}, \boldsymbol{m}) \in (\tilde{\boldsymbol{h}}, \tilde{\boldsymbol{m}}) \oplus S$, only $(\tilde{\boldsymbol{h}}, \tilde{\boldsymbol{m}})$ is feasible in (3). So, if $(\tilde{\boldsymbol{h}}, \tilde{\boldsymbol{m}})$ solves (10) then $(\tilde{\boldsymbol{h}}, \tilde{\boldsymbol{m}})$ solves (3). $\qquad\square$

## A.2   Proof of Lemma 2:

Define a one-sided loss function:

$$\mathcal{L}(\boldsymbol{h}, \boldsymbol{m}) := \tfrac{1}{L} \sum_{\ell=1}^{L} \left[ 2|y_\ell| - s_\ell \boldsymbol{c}_\ell^\mathsf{T} \tilde{\boldsymbol{m}} \boldsymbol{b}_\ell^\mathsf{T} \boldsymbol{h} - s_\ell \boldsymbol{b}_\ell^\mathsf{T} \tilde{\boldsymbol{h}} \boldsymbol{c}_\ell^\mathsf{T} \boldsymbol{m} \right]_+,$$

where $(\cdot)_+$ denotes the positive side. The LP in (10) can now be equivalently expressed as

$$(\hat{\boldsymbol{h}}, \hat{\boldsymbol{m}}) := \operatorname*{axrgmin}_{(\boldsymbol{h}, \boldsymbol{m}) \in \mathbb{R}^{K+N}} \|\boldsymbol{h}\|_1 + \|\boldsymbol{m}\|_1 \text{ subject to } \mathcal{L}(\boldsymbol{h}, \boldsymbol{m}) \leq 0. \tag{16}$$

We want to show that there is no feasible descent direction $(\delta \boldsymbol{h}, \delta \boldsymbol{m}) \in \mathcal{D}$ around the true solution $(\tilde{\boldsymbol{h}}, \tilde{\boldsymbol{m}})$. Since $(\delta \boldsymbol{h}, \delta \boldsymbol{m})$ is a feasible perturbation from the proposed optimal $(\tilde{\boldsymbol{h}}, \tilde{\boldsymbol{m}})$, we have from (16)

$$\mathcal{L}(\tilde{\boldsymbol{h}} + \delta \boldsymbol{h}, \tilde{\boldsymbol{m}} + \delta \boldsymbol{m}) \leq 0. \tag{17}$$

We begin by expanding the loss function $\mathcal{L}(\tilde{\boldsymbol{h}} + \delta \boldsymbol{h}, \tilde{\boldsymbol{m}} + \delta \boldsymbol{m})$ below

$$\mathcal{L}(\tilde{\boldsymbol{h}} + \delta \boldsymbol{h}, \tilde{\boldsymbol{m}} + \delta \boldsymbol{m}) = \tfrac{1}{L} \sum_{\ell=1}^{L} \left[ s_\ell (2y_\ell - \boldsymbol{b}_\ell^\mathsf{T} \tilde{\boldsymbol{h}} \boldsymbol{c}_\ell^\mathsf{T} (\tilde{\boldsymbol{m}} + \delta \boldsymbol{m}) - \boldsymbol{c}_\ell^\mathsf{T} \tilde{\boldsymbol{m}} \boldsymbol{b}_\ell^\mathsf{T} (\tilde{\boldsymbol{h}} + \delta \boldsymbol{h}) \right]_+$$

$$\geq \tfrac{1}{L} \sum_{\ell=1}^{L} \left[ - s_\ell \boldsymbol{b}_\ell^\mathsf{T} \tilde{\boldsymbol{h}} \boldsymbol{c}_\ell^\mathsf{T} \delta \boldsymbol{m} - s_\ell \boldsymbol{c}_\ell^\mathsf{T} \tilde{\boldsymbol{m}} \boldsymbol{b}_\ell^\mathsf{T} \delta \boldsymbol{h} \right]_+. \tag{18}$$

Let $\psi_t(s) := (s)_+ - (s - t)_+$. Using the fact that $\psi_t(s) \leq (s)_+$, and that for every $\alpha, t \geq 0$, and $s \in \mathbb{R}$, $\psi_{\alpha t}(s) = t \psi_\alpha(\tfrac{s}{t})$, we have

$$\tfrac{1}{L} \sum_{\ell=1}^{L} \left[ - s_\ell \boldsymbol{b}_\ell^\mathsf{T} \tilde{\boldsymbol{h}} \boldsymbol{c}_\ell^\mathsf{T} \delta \boldsymbol{m} - s_\ell \boldsymbol{c}_\ell^\mathsf{T} \tilde{\boldsymbol{m}} \boldsymbol{b}_\ell^\mathsf{T} \delta \boldsymbol{h} \right]_+ \geq \tfrac{1}{L} \sum_{\ell=1}^{L} \psi_{\tau \|(\delta \boldsymbol{h}, \delta \boldsymbol{m})\|_2} \left( - s_\ell \boldsymbol{b}_\ell^\mathsf{T} \tilde{\boldsymbol{h}} \boldsymbol{c}_\ell^\mathsf{T} \delta \boldsymbol{m} - s_\ell \boldsymbol{c}_\ell^\mathsf{T} \tilde{\boldsymbol{m}} \boldsymbol{b}_\ell^\mathsf{T} \delta \boldsymbol{h} \right)$$

$$= \|(\delta \boldsymbol{h}, \delta \boldsymbol{m})\|_2 \cdot \tfrac{1}{L} \sum_{\ell=1}^{L} \psi_\tau \left( - s_\ell \left\langle (\boldsymbol{c}_\ell^\mathsf{T} \tilde{\boldsymbol{m}} \boldsymbol{b}_\ell, \boldsymbol{b}_\ell^\mathsf{T} \tilde{\boldsymbol{h}} \boldsymbol{c}_\ell), \tfrac{(\delta \boldsymbol{h}, \delta \boldsymbol{m})}{\|(\delta \boldsymbol{h}, \delta \boldsymbol{m})\|_2} \right\rangle \right)$$

$$= \|(\delta \boldsymbol{h}, \delta \boldsymbol{m})\|_2 \left[ \tfrac{1}{L} \sum_{\ell=1}^{L} \mathrm{E}\, \psi_\tau \left( - s_\ell \left\langle (\boldsymbol{c}_\ell^\mathsf{T} \tilde{\boldsymbol{m}} \boldsymbol{b}_\ell, \boldsymbol{b}_\ell^\mathsf{T} \tilde{\boldsymbol{h}} \boldsymbol{c}_\ell), \tfrac{(\delta \boldsymbol{h}, \delta \boldsymbol{m})}{\|(\delta \boldsymbol{h}, \delta \boldsymbol{m})\|_2} \right\rangle \right) - \right.$$

$$\tfrac{1}{L} \sum_{\ell=1}^{L} \left( \mathrm{E}\, \psi_\tau \left( - s_\ell \left\langle (\boldsymbol{c}_\ell^\mathsf{T} \tilde{\boldsymbol{m}} \boldsymbol{b}_\ell, \boldsymbol{b}_\ell^\mathsf{T} \tilde{\boldsymbol{h}} \boldsymbol{c}_\ell), \tfrac{(\delta \boldsymbol{h}, \delta \boldsymbol{m})}{\|(\delta \boldsymbol{h}, \delta \boldsymbol{m})\|_2} \right\rangle \right) - \psi_\tau \left( - s_\ell \left\langle (\boldsymbol{c}_\ell^\mathsf{T} \tilde{\boldsymbol{m}} \boldsymbol{b}_\ell, \boldsymbol{b}_\ell^\mathsf{T} \tilde{\boldsymbol{h}} \boldsymbol{c}_\ell), \tfrac{(\delta \boldsymbol{h}, \delta \boldsymbol{m})}{\|(\delta \boldsymbol{h}, \delta \boldsymbol{m})\|_2} \right\rangle \right) \right) \right].$$

$$\tag{19}$$

The proof mainly relies on lower bounding the right hand side above uniformly over all $(\delta\boldsymbol{h}, \delta\boldsymbol{m}) \in \mathcal{D}$. To this end, define a centered random process $\mathcal{R}(\boldsymbol{B}, \boldsymbol{C})$ as follows

$$\mathcal{R}(\boldsymbol{B}, \boldsymbol{C}) := \sup_{(\delta\boldsymbol{h}, \delta\boldsymbol{m}) \in \mathcal{D}} \frac{1}{L} \sum_{\ell=1}^{L} \left[ \mathrm{E}\,\psi_\tau \left( -s_\ell \left\langle (\boldsymbol{c}_\ell^\mathsf{T} \tilde{\boldsymbol{m}} \boldsymbol{b}_\ell, \boldsymbol{b}_\ell^\mathsf{T} \tilde{\boldsymbol{h}} \boldsymbol{c}_\ell), \frac{(\delta\boldsymbol{h}, \delta\boldsymbol{m})}{\|(\delta\boldsymbol{h}, \delta\boldsymbol{m})\|_2} \right\rangle \right) \right.$$
$$\left. - \psi_\tau \left( -s_\ell \left\langle (\boldsymbol{c}_\ell^\mathsf{T} \tilde{\boldsymbol{m}} \boldsymbol{b}_\ell, \boldsymbol{b}_\ell^\mathsf{T} \tilde{\boldsymbol{h}} \boldsymbol{c}_\ell), \frac{(\delta\boldsymbol{h}, \delta\boldsymbol{m})}{\|(\delta\boldsymbol{h}, \delta\boldsymbol{m})\|_2} \right\rangle \right) \right],$$

and an application of bounded difference inequality McDiarmid [1989] yields that $\mathcal{R}(\boldsymbol{B}, \boldsymbol{C}) \leq \mathrm{E}\,\mathcal{R}(\boldsymbol{B}, \boldsymbol{C}) + t\tau/\sqrt{L}$ with probability at least $1 - \mathrm{e}^{-2Lt^2}$. It remains to evaluate $\mathrm{E}\,\mathcal{R}(\boldsymbol{B}, \boldsymbol{C})$, which after using a simple symmetrization inequality van der Vaart and Wellner [1997] yields

$$\mathrm{E}\,\mathcal{R}(\boldsymbol{B}, \boldsymbol{C}) \leq 2\,\mathrm{E} \sup_{(\delta\boldsymbol{h}, \delta\boldsymbol{m}) \in \mathcal{D} \cap \mathcal{B}} \frac{1}{L} \sum_{\ell=1}^{L} \varepsilon_\ell \psi_\tau \left( -s_\ell \left\langle (\boldsymbol{c}_\ell^\mathsf{T} \tilde{\boldsymbol{m}} \boldsymbol{b}_\ell, \boldsymbol{b}_\ell^\mathsf{T} \tilde{\boldsymbol{h}} \boldsymbol{c}_\ell), \frac{(\delta\boldsymbol{h}, \delta\boldsymbol{m})}{\|(\delta\boldsymbol{h}, \delta\boldsymbol{m})\|_2} \right\rangle \right), \quad (20)$$

where $\varepsilon_1, \varepsilon_2, \ldots, \varepsilon_L$ are independent Rademacher random variables. Using the fact that $\psi_t(s)$ is a contraction: $|\psi_t(\alpha_1) - \psi_t(\alpha_2)| \leq |\alpha_1 - \alpha_2|$ for all $\alpha_1, \alpha_2 \in \mathbb{R}$, we have from the Rademacher contraction inequality Ledoux and Talagrand [2013] that

$$\mathrm{E} \sup_{(\delta\boldsymbol{h}, \delta\boldsymbol{m}) \in \mathcal{D}} \frac{1}{L} \sum_{\ell=1}^{L} \varepsilon_\ell \psi_\tau \left( -s_\ell \left\langle (\boldsymbol{c}_\ell^\mathsf{T} \tilde{\boldsymbol{m}} \boldsymbol{b}_\ell, \boldsymbol{b}_\ell^\mathsf{T} \tilde{\boldsymbol{h}} \boldsymbol{c}_\ell), \frac{(\delta\boldsymbol{h}, \delta\boldsymbol{m})}{\|(\delta\boldsymbol{h}, \delta\boldsymbol{m})\|_2} \right\rangle \right)$$
$$\leq \mathrm{E} \sup_{(\delta\boldsymbol{h}, \delta\boldsymbol{m}) \in \mathcal{D}} \frac{1}{L} \sum_{\ell=1}^{L} -\varepsilon_\ell s_\ell \left\langle (\boldsymbol{c}_\ell^\mathsf{T} \tilde{\boldsymbol{m}} \boldsymbol{b}_\ell, \boldsymbol{b}_\ell^\mathsf{T} \tilde{\boldsymbol{h}} \boldsymbol{c}_\ell), \frac{(\delta\boldsymbol{h}, \delta\boldsymbol{m})}{\|(\delta\boldsymbol{h}, \delta\boldsymbol{m})\|_2} \right\rangle$$
$$= \mathrm{E} \sup_{(\delta\boldsymbol{h}, \delta\boldsymbol{m}) \in \mathcal{D}} \frac{1}{L} \sum_{\ell=1}^{L} \varepsilon_\ell \left\langle (\boldsymbol{c}_\ell^\mathsf{T} \tilde{\boldsymbol{m}} \boldsymbol{b}_\ell, \boldsymbol{b}_\ell^\mathsf{T} \tilde{\boldsymbol{h}} \boldsymbol{c}_\ell), \frac{(\delta\boldsymbol{h}, \delta\boldsymbol{m})}{\|(\delta\boldsymbol{h}, \delta\boldsymbol{m})\|_2} \right\rangle, \quad (21)$$

where the last equality is the result of the fact that multiplying Rademacher random variables with signs does not change the distribution. In addition, using the facts that $t\mathbf{1}(s \geq t) \leq \psi_t(s)$, and that random vectors $\{(\boldsymbol{c}_\ell^\mathsf{T} \tilde{\boldsymbol{m}} \boldsymbol{b}_\ell, \boldsymbol{b}_\ell^\mathsf{T} \tilde{\boldsymbol{h}} \boldsymbol{c}_\ell)\}_{\ell=1}^{L}$ are identically distributed and the distribution is symmetric, it follows

$$\tau \mathbb{P} \left( -s_\ell \left\langle (\boldsymbol{c}_\ell^\mathsf{T} \tilde{\boldsymbol{m}} \boldsymbol{b}_\ell, \boldsymbol{b}_\ell^\mathsf{T} \tilde{\boldsymbol{h}} \boldsymbol{c}_\ell), \frac{(\delta\boldsymbol{h}, \delta\boldsymbol{m})}{\|(\delta\boldsymbol{h}, \delta\boldsymbol{m})\|_2} \right\rangle \geq \tau \right) = \tau \mathbb{P} \left( \left\langle (\boldsymbol{c}_\ell^\mathsf{T} \tilde{\boldsymbol{m}} \boldsymbol{b}_\ell, \boldsymbol{b}_\ell^\mathsf{T} \tilde{\boldsymbol{h}} \boldsymbol{c}_\ell), \frac{(\delta\boldsymbol{h}, \delta\boldsymbol{m})}{\|(\delta\boldsymbol{h}, \delta\boldsymbol{m})\|_2} \right\rangle \geq \tau \right)$$
$$= \tau \mathrm{E} \left[ \mathbf{1} \left( \left\langle (\boldsymbol{c}_\ell^\mathsf{T} \tilde{\boldsymbol{m}} \boldsymbol{b}_\ell, \boldsymbol{b}_\ell^\mathsf{T} \tilde{\boldsymbol{h}} \boldsymbol{c}_\ell), \frac{(\delta\boldsymbol{h}, \delta\boldsymbol{m})}{\|(\delta\boldsymbol{h}, \delta\boldsymbol{m})\|_2} \right\rangle \geq \tau \right) \right] \leq \mathrm{E}\,\psi_\tau \left( \left\langle (\boldsymbol{c}_\ell^\mathsf{T} \tilde{\boldsymbol{m}} \boldsymbol{b}_\ell, \boldsymbol{b}_\ell^\mathsf{T} \tilde{\boldsymbol{h}} \boldsymbol{c}_\ell), \frac{(\delta\boldsymbol{h}, \delta\boldsymbol{m})}{\|(\delta\boldsymbol{h}, \delta\boldsymbol{m})\|_2} \right\rangle \right). \quad (22)$$

Plugging (22), and (21) in (19), we have

$$\frac{1}{L} \sum_{\ell=1}^{L} \left[ -s_\ell \left\langle (\boldsymbol{c}_\ell^\mathsf{T} \tilde{\boldsymbol{m}} \boldsymbol{b}_\ell, \boldsymbol{b}_\ell^\mathsf{T} \tilde{\boldsymbol{h}} \boldsymbol{c}_\ell), \frac{(\delta\boldsymbol{h}, \delta\boldsymbol{m})}{\|(\delta\boldsymbol{h}, \delta\boldsymbol{m})\|_2} \right\rangle \right]_+ \geq$$
$$\tau \|(\delta\boldsymbol{h}, \delta\boldsymbol{m})\|_2 \mathbb{P} \left( \left\langle (\boldsymbol{c}_\ell^\mathsf{T} \tilde{\boldsymbol{m}} \boldsymbol{b}_\ell, \boldsymbol{b}_\ell^\mathsf{T} \tilde{\boldsymbol{h}} \boldsymbol{c}_\ell), \frac{(\delta\boldsymbol{h}, \delta\boldsymbol{m})}{\|(\delta\boldsymbol{h}, \delta\boldsymbol{m})\|_2} \right\rangle \geq \tau \right)$$
$$- \|(\delta\boldsymbol{h}, \delta\boldsymbol{m})\|_2 \left( 2\,\mathrm{E} \sup_{(\delta\boldsymbol{h}, \delta\boldsymbol{m}) \in \mathcal{D}} \frac{1}{L} \sum_{\ell=1}^{L} \varepsilon_\ell \left\langle (\boldsymbol{c}_\ell^\mathsf{T} \tilde{\boldsymbol{m}} \boldsymbol{b}_\ell, \boldsymbol{b}_\ell^\mathsf{T} \tilde{\boldsymbol{h}} \boldsymbol{c}_\ell), \frac{(\delta\boldsymbol{h}, \delta\boldsymbol{m})}{\|(\delta\boldsymbol{h}, \delta\boldsymbol{m})\|_2} \right\rangle + \frac{t\tau}{\sqrt{L}} \right)$$

Combining this with (17) and (18), we obtain the final result

$$\|(\delta\boldsymbol{h}, \delta\boldsymbol{m})\|_2 \left[ \tau \mathbb{P} \left( \left\langle (\boldsymbol{c}_\ell^\mathsf{T} \tilde{\boldsymbol{m}} \boldsymbol{b}_\ell, \boldsymbol{b}_\ell^\mathsf{T} \tilde{\boldsymbol{h}} \boldsymbol{c}_\ell), \frac{(\delta\boldsymbol{h}, \delta\boldsymbol{m})}{\|(\delta\boldsymbol{h}, \delta\boldsymbol{m})\|_2} \right\rangle \geq \tau \right) \right.$$
$$\left. - \left( 2\,\mathrm{E} \sup_{(\delta\boldsymbol{h}, \delta\boldsymbol{m}) \in \mathcal{D}} \frac{1}{L} \sum_{\ell=1}^{L} \varepsilon_\ell \left\langle (\boldsymbol{c}_\ell^\mathsf{T} \tilde{\boldsymbol{m}} \boldsymbol{b}_\ell, \boldsymbol{b}_\ell^\mathsf{T} \tilde{\boldsymbol{h}} \boldsymbol{c}_\ell), \frac{(\delta\boldsymbol{h}, \delta\boldsymbol{m})}{\|(\delta\boldsymbol{h}, \delta\boldsymbol{m})\|_2} \right\rangle + \frac{t\tau}{\sqrt{L}} \right) \right] \leq 0.$$

Using the definitions in (13), and (14), we can write

$$\|(\delta\boldsymbol{h}, \delta\boldsymbol{m})\|_2 \left( \tau \mathfrak{p}_\tau(\mathcal{D}) - \frac{(2\mathfrak{C}(\mathcal{D}) + t\tau)}{\sqrt{L}} \right) \leq 0.$$

It is clear that choosing $L \geq \left( \frac{2\mathfrak{C}(\mathcal{D}) + t\tau}{\tau \mathfrak{p}_\tau(\mathcal{D})} \right)^2$ implies

$$\|(\delta h, \delta m)\|_2 \leq 0,$$

which directly means that $(\delta h, \delta m) = (0, 0)$. Recall that $\mathcal{S} \subset \mathcal{N}$, and $\mathcal{D} \perp \mathcal{N}$, where $\mathcal{S}$ is defined in (11), this implies that the minimizer $(\hat{h}, \hat{m})$ of the LP (10) resides in the set $(\tilde{h}, \tilde{m}) \oplus \mathcal{S}$. This completes the proof of Lemma 2.

## A.3 Proof of Theorem 2:

In light of Lemma 2, the proof of Theorem 2 comes down to computing the Rademacher complexity $\mathfrak{C}(\mathcal{D})$ defined in (13), and the tail probability estimate $\mathfrak{p}_\tau(\mathcal{D})$ defined in (14) of the set of descent directions $\mathcal{D}$ defined in (12).

**Upper Bound on Rademacher Complexity:** We will start by evaluating $\mathfrak{C}(\mathcal{D})$

$$
\begin{aligned}
\mathfrak{C}(\mathcal{D}) &= \mathrm{E} \sup_{(\delta h, \delta m) \in \mathcal{D}} \frac{1}{\sqrt{L}} \sum_{\ell=1}^{L} \varepsilon_\ell \left\langle (c_\ell^\mathsf{T} \tilde{m} b_\ell, b_\ell^\mathsf{T} \tilde{h} c_\ell), \frac{(\delta h, \delta m)}{\|(\delta h, \delta m)\|_2} \right\rangle \\
&\leq \mathrm{E} \left\| \frac{1}{\sqrt{L}} \sum_{\ell=1}^{L} \varepsilon_\ell \left( c_\ell^\mathsf{T} \tilde{m} b_\ell|_{\Gamma_h}, b_\ell^\mathsf{T} \tilde{h} c_\ell|_{\Gamma_m} \right) \right\|_2 \cdot \sup_{(\delta h, \delta m) \in \mathcal{D}} \left\| \frac{(\delta h_{\Gamma_h}, \delta m_{\Gamma_m})}{\|(\delta h, \delta m)\|_2} \right\|_2 \\
&\quad + \mathrm{E} \left\| \frac{1}{\sqrt{L}} \sum_{\ell=1}^{L} \varepsilon_\ell \left( c_\ell^\mathsf{T} \tilde{m} b_\ell|_{\Gamma_h^c}, b_\ell^\mathsf{T} \tilde{h} c_\ell|_{\Gamma_m^c} \right) \right\|_\infty \cdot \sup_{(\delta h, \delta m) \in \mathcal{D}} \left\| \frac{(\delta h_{\Gamma_h^c}, \delta m_{\Gamma_m^c})}{\|(\delta h, \delta m)\|_2} \right\|_1.
\end{aligned}
\tag{23}
$$

First note that on set $\mathcal{D}$ (12), we have

$$\left\| \frac{(\delta h_{\Gamma_h^c}, \delta m_{\Gamma_m^c})}{\|(\delta h, \delta m)\|_2} \right\|_1 \leq \sqrt{S_1 + S_2} \left\| \frac{(\delta h_{\Gamma_h}, \delta m_{\Gamma_m})}{\|(\delta h, \delta m)\|_2} \right\|_2 \leq \sqrt{S_1 + S_2}.$$

As for the remaining terms, we begin by writing

$$
\begin{aligned}
\mathrm{E} \left\| \frac{1}{\sqrt{L}} \sum_{\ell=1}^{L} \varepsilon_\ell \left( c_\ell^\mathsf{T} \tilde{m} b_\ell|_{\Gamma_h}, b_\ell^\mathsf{T} \tilde{h} c_\ell|_{\Gamma_m} \right) \right\|_2 &\leq \sqrt{\mathrm{E} \left\| \frac{1}{\sqrt{L}} \sum_{\ell=1}^{L} \varepsilon_\ell \left( c_\ell^\mathsf{T} \tilde{m} b_\ell|_{\Gamma_h}, b_\ell^\mathsf{T} \tilde{h} c_\ell|_{\Gamma_m} \right) \right\|_2^2} \\
&= \sqrt{\frac{1}{L} \sum_{\ell=1}^{L} \mathrm{E} \left( |c_\ell^\mathsf{T} \tilde{m}|^2 \|b_\ell^\mathsf{T}|_{\Gamma_h}\|_2^2 + |b_\ell \tilde{h}|^2 \|c_\ell|_{\Gamma_m}\|_2^2 \right)} \\
&= \sqrt{\|\tilde{m}\|_2^2 S_1 + \|\tilde{h}\|_2^2 S_2},
\end{aligned}
$$

and the second term in (23) is

$$
\begin{aligned}
\mathrm{E} \left\| \frac{1}{\sqrt{L}} \sum_{\ell=1}^{L} \varepsilon_\ell (b_\ell^\mathsf{T} \tilde{h} c_\ell|_{\Gamma_m^c}, c_\ell^\mathsf{T} \tilde{m} b_\ell|_{\Gamma_h^c}) \right\|_\infty &\leq \sqrt{\mathrm{E} \left\| \frac{1}{\sqrt{L}} \sum_{\ell=1}^{L} \varepsilon_\ell (b_\ell^\mathsf{T} \tilde{h} c_\ell|_{\Gamma_m^c}, c_\ell^\mathsf{T} \tilde{m} b_\ell|_{\Gamma_h^c}) \right\|_\infty^2} \\
&\leq \sqrt{2e \log(K+N) \cdot \frac{1}{L} \sum_{\ell=1}^{L} \mathrm{E} \max \left\{ |c_\ell^\mathsf{T} \tilde{m}|^2 \|b_\ell|_{\Gamma_h^c}\|_\infty^2, |b_\ell^\mathsf{T} \tilde{h}|^2 \|c_\ell|_{\Gamma_m^c}\|_\infty^2 \right\}} \\
&\leq \sqrt{2e \log(K+N) \, \mathrm{E} \max\{ |b^\mathsf{T} \tilde{h}|^2 \|c|_{\Gamma_m^c}\|_\infty^2, |c^\mathsf{T} \tilde{m}|^2 \|b|_{\Gamma_h^c}\|_\infty^2 \}} \\
&\leq C \sqrt{\max\{\|\tilde{h}\|_2^2, \|\tilde{m}\|_2^2\} \log^2(K+N)},
\end{aligned}
$$

where the second inequality by the application of Lemma 5.2.2 in Akritas et al. [2016], and the final equality is due to the fact that $\|c|_{\Gamma_m^c}\|_\infty^2$, and $\|b|_{\Gamma_h^c}\|_\infty^2$ are subexponential and using Lemma 3 in van de Geer and Lederer [2013].

Plugging the bounds above back in (23), we obtain the upper bound on the Rademacher complexity given below

$$\mathfrak{C}(\mathcal{D}) \leq C \sqrt{\left( \|\tilde{m}\|_2^2 + \|\tilde{h}\|_2^2 \right)(S_1 + S_2) \log^2(K+N)}. \tag{24}$$

**Tail Probability:** To apply the result in Lemma 2, we also need to evaluate

$$\mathfrak{p}_\tau(\mathcal{D}) = \inf_{(\delta \boldsymbol{h}, \delta \boldsymbol{m}) \in \mathcal{D}} \mathbb{P}\left( \left\langle (\boldsymbol{c}_\ell^\mathsf{T} \tilde{\boldsymbol{m}} \boldsymbol{b}_\ell, \boldsymbol{b}_\ell^\mathsf{T} \tilde{\boldsymbol{h}} \boldsymbol{c}_\ell), \frac{(\delta \boldsymbol{h}, \delta \boldsymbol{m})}{\|(\delta \boldsymbol{h}, \delta \boldsymbol{m})\|_2} \right\rangle \geq \tau \right). \tag{25}$$

It suffice to estimate the probability $\mathbb{P}(|\boldsymbol{b}_\ell^\mathsf{T} \tilde{\boldsymbol{h}} \boldsymbol{c}_\ell^\mathsf{T} \delta \boldsymbol{m} + \boldsymbol{b}_\ell^\mathsf{T} \delta \boldsymbol{h} \boldsymbol{c}_\ell^\mathsf{T} \tilde{\boldsymbol{m}}| \geq \tau)$. Using Paley-Zygmund inequality, we obtain

$$\mathbb{P}\left( |\boldsymbol{b}_\ell^\mathsf{T} \tilde{\boldsymbol{h}} \boldsymbol{c}_\ell^\mathsf{T} \delta \boldsymbol{m} + \boldsymbol{b}_\ell^\mathsf{T} \delta \boldsymbol{h} \boldsymbol{c}_\ell^\mathsf{T} \tilde{\boldsymbol{m}}|^2 \geq \frac{1}{2} \operatorname{E} |\boldsymbol{b}_\ell^\mathsf{T} \tilde{\boldsymbol{h}} \boldsymbol{c}_\ell^\mathsf{T} \delta \boldsymbol{m} + \boldsymbol{b}_\ell^\mathsf{T} \delta \boldsymbol{h} \boldsymbol{c}_\ell^\mathsf{T} \tilde{\boldsymbol{m}}|^2 \right)$$

$$\geq \frac{1}{4} \cdot \frac{(\operatorname{E} |\boldsymbol{b}_\ell^\mathsf{T} \tilde{\boldsymbol{h}} \boldsymbol{c}_\ell^\mathsf{T} \delta \boldsymbol{m} + \boldsymbol{b}_\ell^\mathsf{T} \delta \boldsymbol{h} \boldsymbol{c}_\ell^\mathsf{T} \tilde{\boldsymbol{m}}|^2)^2}{\operatorname{E} |\boldsymbol{b}_\ell^\mathsf{T} \tilde{\boldsymbol{h}} \boldsymbol{c}_\ell^\mathsf{T} \delta \boldsymbol{m} + \boldsymbol{b}_\ell^\mathsf{T} \delta \boldsymbol{h} \boldsymbol{c}_\ell^\mathsf{T} \tilde{\boldsymbol{m}}|^4}.$$

By the norm equivalence of Gaussian random variables, we have that $(\operatorname{E} |\boldsymbol{b}_\ell^\mathsf{T} \tilde{\boldsymbol{h}} \boldsymbol{c}_\ell^\mathsf{T} \delta \boldsymbol{m} + \boldsymbol{b}_\ell^\mathsf{T} \delta \boldsymbol{h} \boldsymbol{c}_\ell^\mathsf{T} \tilde{\boldsymbol{m}}|^4)^{1/4} \leq c(\operatorname{E} |\boldsymbol{b}_\ell^\mathsf{T} \tilde{\boldsymbol{h}} \boldsymbol{c}_\ell^\mathsf{T} \delta \boldsymbol{m} + \boldsymbol{b}_\ell^\mathsf{T} \delta \boldsymbol{h} \boldsymbol{c}_\ell^\mathsf{T} \tilde{\boldsymbol{m}}|^2)^{1/2}$, this implies that

$$\mathbb{P}(|\boldsymbol{b}_\ell^\mathsf{T} \tilde{\boldsymbol{h}} \boldsymbol{c}_\ell^\mathsf{T} \delta \boldsymbol{m} + \boldsymbol{b}_\ell^\mathsf{T} \delta \boldsymbol{h} \boldsymbol{c}_\ell^\mathsf{T} \tilde{\boldsymbol{m}}|^2 \geq \frac{1}{2} \operatorname{E} |\boldsymbol{b}_\ell^\mathsf{T} \tilde{\boldsymbol{h}} \boldsymbol{c}_\ell^\mathsf{T} \delta \boldsymbol{m} + \boldsymbol{b}_\ell^\mathsf{T} \delta \boldsymbol{h} \boldsymbol{c}_\ell^\mathsf{T} \tilde{\boldsymbol{m}}|^2) \geq \frac{1}{4} \cdot \frac{1}{c^4}. \tag{26}$$

Finally, a simple calculation shows that $\operatorname{E} |\boldsymbol{b}_\ell^\mathsf{T} \tilde{\boldsymbol{h}} \boldsymbol{c}_\ell^\mathsf{T} \delta \boldsymbol{m} + \boldsymbol{b}_\ell^\mathsf{T} \delta \boldsymbol{h} \boldsymbol{c}_\ell^\mathsf{T} \tilde{\boldsymbol{m}}|^2 \geq \min\{\|\tilde{\boldsymbol{h}}\|_2^2, \|\tilde{\boldsymbol{m}}\|_2^2\}$ $\left( \|\delta \boldsymbol{m}\|_2^2 + \|\delta \boldsymbol{h}\|_2^2 \right)$.

$$\operatorname{E} |\boldsymbol{b}_\ell^\mathsf{T} \tilde{\boldsymbol{h}} \boldsymbol{c}_\ell^\mathsf{T} \delta \boldsymbol{m} + \boldsymbol{b}_\ell^\mathsf{T} \delta \boldsymbol{h} \boldsymbol{c}_\ell^\mathsf{T} \tilde{\boldsymbol{m}}|^2 = \operatorname{E}_{\boldsymbol{b}} \operatorname{E}_{\boldsymbol{c}} \tilde{\boldsymbol{h}}^\top \boldsymbol{b}_\ell \boldsymbol{b}_\ell^\mathsf{T} \tilde{\boldsymbol{h}} \delta \boldsymbol{m}^\top \boldsymbol{c}_\ell \boldsymbol{c}_\ell^\mathsf{T} \delta \boldsymbol{m} + \delta \boldsymbol{h}^\top \boldsymbol{b}_\ell \boldsymbol{b}_\ell^\mathsf{T} \delta \boldsymbol{h} \tilde{\boldsymbol{m}}^\top \boldsymbol{c}_\ell \boldsymbol{c}_\ell^\mathsf{T} \tilde{\boldsymbol{m}}$$
$$+ 2 \operatorname{E}_{\boldsymbol{b}} \operatorname{E}_{\boldsymbol{c}} \delta \boldsymbol{h}^\top \boldsymbol{b}_\ell \boldsymbol{b}_\ell^\mathsf{T} \tilde{\boldsymbol{h}} \delta \boldsymbol{m}^\top \boldsymbol{c}_\ell \boldsymbol{c}_\ell^\mathsf{T} \tilde{\boldsymbol{m}}$$
$$= \operatorname{E}_{\boldsymbol{b}}(\|\delta \boldsymbol{m}\|_2^2 \tilde{\boldsymbol{h}}^\top \boldsymbol{b}_\ell \boldsymbol{b}_\ell^\mathsf{T} \tilde{\boldsymbol{h}} + \|\tilde{\boldsymbol{m}}\|_2^2 \delta \boldsymbol{h}^\top \boldsymbol{b}_\ell \boldsymbol{b}_\ell^\mathsf{T} \delta \boldsymbol{h} + 2 \delta \boldsymbol{m}^\top \tilde{\boldsymbol{m}} \delta \boldsymbol{h}^\top \boldsymbol{b}_\ell \boldsymbol{b}_\ell^\mathsf{T} \tilde{\boldsymbol{h}})$$
$$= \|\delta \boldsymbol{m}\|_2^2 \|\tilde{\boldsymbol{h}}\|_2^2 + \|\tilde{\boldsymbol{m}}\|_2^2 \|\delta \boldsymbol{h}\|_2^2 + 2 \delta \boldsymbol{m}^\top \tilde{\boldsymbol{m}} \delta \boldsymbol{h}^\top \tilde{\boldsymbol{h}}$$
$$= \|\delta \boldsymbol{m}\|_2^2 \|\tilde{\boldsymbol{h}}\|_2^2 + \|\tilde{\boldsymbol{m}}\|_2^2 \|\delta \boldsymbol{h}\|_2^2 + 2 (\delta \boldsymbol{h}^\top \tilde{\boldsymbol{h}})^2$$
$$\geq \|\delta \boldsymbol{m}\|_2^2 \|\tilde{\boldsymbol{h}}\|_2^2 + \|\tilde{\boldsymbol{m}}\|_2^2 \|\delta \boldsymbol{h}\|_2^2,$$
$$\geq \min\{\|\tilde{\boldsymbol{h}}\|_2^2, \|\tilde{\boldsymbol{m}}\|_2^2\} \left( \|\delta \boldsymbol{m}\|_2^2 + \|\delta \boldsymbol{h}\|_2^2 \right),$$

where the last equality follows using the fact $(\delta \boldsymbol{h}, \delta \boldsymbol{m}) \in \mathcal{D} \subset \mathcal{N}_\perp$, and hence $\mathcal{D} \perp \mathcal{N}$, which implies that $\delta \boldsymbol{h}^\top \tilde{\boldsymbol{h}} = \delta \boldsymbol{m}^\top \tilde{\boldsymbol{m}}$. Normalizing by $\|(\delta \boldsymbol{h}, \delta \boldsymbol{m})\|_2$, and comparing with (25) directly shows that $\tau^2 = \min\{\|\tilde{\boldsymbol{h}}\|_2^2, \|\tilde{\boldsymbol{m}}\|_2^2\}$, and $\mathfrak{p}_\tau(\mathcal{D}) \geq \frac{1}{8c^4}$. Plugging these results and the Rademacher complexing bound in (24) in Lemma 2 proves Theorem 2. $\qquad \square$

### A.4 Evaluation of the Projection Operator

Given a point $(\boldsymbol{x}', \boldsymbol{w}', \boldsymbol{\xi}') \in \mathbb{R}^{3L}$, in this section we focus on deriving a closed-form expression for $\operatorname{proj}_{\mathcal{C}}\left( (\boldsymbol{x}', \boldsymbol{w}', \boldsymbol{\xi}') \right)$, where

$$\mathcal{C} = \left\{ (\boldsymbol{x}, \boldsymbol{w}, \boldsymbol{\xi}) \in \mathbb{R}^{3L} \mid s_\ell(\xi_\ell + x_\ell) w_\ell \geq |y_\ell|, \; t_\ell w_\ell \geq 0, \; \ell = 1, \dots, L \right\}$$

is the convex feasible set of (6). It is straightforward to see that the resulting projection program decouples into $L$ convex programs in $\mathbb{R}^3$ as

$$\arg\min_{x \in \mathbb{R}, w \in \mathbb{R}, \xi \in \mathbb{R}} \frac{1}{2} \left\| \begin{pmatrix} x \\ w \\ \xi \end{pmatrix} - \begin{pmatrix} x'_\ell \\ w'_\ell \\ \xi'_\ell \end{pmatrix} \right\|_2^2 \quad s.t. \; |y_\ell| - s_\ell x w - s_\ell \xi w \leq 0, \quad -t_\ell w \leq 0. \tag{27}$$

Throughout this derivation we assume that $|y_\ell| > 0$ (derivation of the projection for the case $y_\ell$ is easy) and as a result of which the second constraint $-t_\ell w \leq 0$ is never active (because then $w = 0$ and the first constraint requires that $|y_\ell| \leq 0$). We also consistently use the fact that $t_\ell$ and $s_\ell$ are signs and nonzero.

Forming the Lagrangian as

$$\mathcal{L}(x, w, \xi, \mu_1, \mu_2) = \frac{1}{2} \left\| \begin{pmatrix} x \\ w \\ \xi \end{pmatrix} - \begin{pmatrix} x'_\ell \\ w'_\ell \\ \xi'_\ell \end{pmatrix} \right\|_2^2 + \mu_1 \left( |y_\ell| - s_\ell x w - s_\ell \xi w \right) - \mu_2 \left( t_\ell w \right),$$

along with the primal constraints, the KKT optimality conditions are

$$\frac{\partial \mathcal{L}}{\partial x} = x - x'_\ell - \mu_1 s_\ell w = 0, \tag{28}$$

$$\frac{\partial \mathcal{L}}{\partial w} = w - w'_\ell - \mu_1 s_\ell x - \mu_1 s_\ell \xi - \mu_2 t_\ell = 0, \tag{29}$$

$$\frac{\partial \mathcal{L}}{\partial \xi} = \xi - \xi'_\ell - \mu_1 s_\ell w = 0, \tag{30}$$

$$\mu_1 \geq 0, \quad \mu_1 \left( |y_\ell| - s_\ell x w - s_\ell \xi w \right) = 0, \tag{31}$$

$$\mu_2 \geq 0, \quad \mu_2 \left( t_\ell w \right) = 0. \tag{32}$$

We now proceed with the possible cases.

**Case 1.** $\mu_1 = \mu_2 = 0$:
In this case we have $(x, w, \xi) = (x'_\ell, w'_\ell, \xi'_\ell)$ and this result would only be acceptable when $|y_\ell| - s_\ell x'_\ell w'_\ell - s_\ell \xi'_\ell w'_\ell \leq 0$ and $t_\ell w'_\ell \geq 0$.

**Case 2.** $\mu_1 = 0, t_\ell w = 0$:
In this case the first feasibility constraint of (27) requires that $|y_\ell| \leq 0$, which is not possible when $|y_\ell| > 0$.

**Case 3.** $|y_\ell| - s_\ell x w - s_\ell \xi w = 0, t_\ell w = 0$:
Similar to the previous case, this cannot happen when $|y_\ell| > 0$.

**Case 4.** $\mu_2 = 0, |y_\ell| - s_\ell x w - s_\ell \xi w = 0$:
In this case we have

$$|y_\ell| = s_\ell x w + s_\ell \xi w.$$

Now combining this observation with (28) and (30) yields

$$|y_\ell| = s_\ell \left( x'_\ell + \mu_1 s_\ell w \right) w + s_\ell \left( \xi'_\ell + \mu_1 s_\ell w \right) w, \tag{33}$$

and therefore

$$\mu_1 = \frac{|y_\ell| - s_\ell \left( x'_\ell + \xi'_\ell \right) w}{2w^2}. \tag{34}$$

Similarly, (29) yields

$$w = w'_\ell + \mu_1 s_\ell \left( x'_\ell + \mu_1 s_\ell w \right) + \mu_1 s_\ell \left( \xi'_\ell + \mu_1 s_\ell w \right). \tag{35}$$

Knowing that $w \neq 0$, $\mu_1$ can be eliminated between (33) and (35) to generate the following forth order polynomial equation in terms of $w$:

$$2w^4 - 2w'_\ell w^3 + s_\ell |y_\ell| \left( x'_\ell + \xi'_\ell \right) w - y_\ell^2 = 0.$$

After solving this 4-th order polynomial equation (e.g., the root command in MATLAB) we pick the real root $w$ which obeys

$$t_\ell w \geq 0, \qquad |y_\ell| - s_\ell \left( x'_\ell + \xi'_\ell \right) w \geq 0. \tag{36}$$

Note that the second inequality in (36) warrants nonnegative values for $\mu_1$ thanks to (34). After picking the right root, we can explicitly obtain $\mu_1$ using (35) and calculate the solutions $x$ and $\xi$ using (28) and (30). Technically, in using the ADMM scheme for each $\ell$ we solve a forth-order polynomial equation and find the projection.