[Reviews · NeurIPS 2018]

Reviewer 1



This paper concerns the problem of recovering two unknown signals from their entrywise product, assuming their signs are known. To constrain the problem further, the two signals are assumed to be sparse with respect to known dictionaries. The entrywise products naturally lead to hyperbola-shaped constraints. The known signs cut the space and help to restrict the feasible set to one branch of hyperbola for each product. This enables convexification of the constraint. Based on this, this paper proposes a convex formulation Eq (2) for the problem. The paper further proposes robust and total-variation extensions of the canonical formulation, and employs the ADMM to solve the resulting general formulation. On the theory side, this paper proves that (Theorem 1) when the dimension of the two vectors is slightly higher in order than the total nonzeros of the hidden sparse vectors (intuitively, the intrinsic degrees of freedom), the proposed convex formulation recovers the two unknown signals, provided the known dictionaries are iid Gaussian. The problem considered in this paper seems new, although there is a similar prior work -- Aghasi, Alireza, Ali Ahmed, and Paul Hand. "BranchHull: Convex bilinear inversion from the entrywise product of signals with known signs." arXiv preprint arXiv:1702.04342 (2017). There are minor differences between the two: 1) For data model, the prior work does not assume sparsity. This causes (simple) changes of the objectives in the convex relaxations, although the constraints are exactly the same. 2) For recovery guarantee, both guarantees are near optimal and requires dimension proportional to the intrinsic degrees of freedom of the signals. The result presented in the current paper implies the prior one. 3) The proof strategies are very similar and there are differences in the technicalities though. So while the results presented are novel, it is also incremental in view of the above points. I think it is worthwhile to clarify the technical novelty in the proof and highlight what significant changes are needed to obtain the current result. Another concern is how relevant is the iid Gaussian assumption on the known dictionaries to practice. Thinking of application such as real-world phase retrieval, one of the dictionaries might be the partial Fourier matrix. Sec 3.2 includes a verification close to this practical situation, although the basis is random partial DCT matrix. I think more practical evaluation is needed. Additional comments: * Line 35: "Unlike prior studies like Ahmed et al. [2014], where the signals are assumed to live in known subspaces, ..." Ahmed et al. [2014] considers the blind convolution problem and so is considerably more different than just without the sparsity constraint. Is the target reference actually Aghasi et al 2016? * Line 39: "Recent work on sparse BIP, specifically sparse blind deconvolution, in Lee et al. [2017] provides ..." looks confusing. Lee et al [2017] considers recovery of sparse rank-1 matrix, and at least superficially it is not about the BIP problem considered here or sparse blind deconvolution. Can the connection be stated more explicitly? Then it would start to make sense to discuss the deficiency of the previous work and hence highlight any improvement the current work can offer. Unless this can be clarified and direct comparison makes sense, I think the claim about the contribution in the next paragraph should be revised to avoid confusion. * The bib information of Aghasi et al 2016 and Lee et al. [2017] seems problematic: the arXiv numbers should probably be switched. * Connection to one-bit compressed sensing: as the signs of both unknown signals are known, and the known dictionaries are assumed to be "generic", i.e., Gaussian, there is a superficial connection to one-bit compressed sensing. Please comment on the connection, and discuss what happens if one approaches the problem as two one-bit compressed sensing problems. * Proof of Lemma 1: Why one needs to multiply |y_\ell| to both sides of the inequalities above Line 340? This does not seem necessary.

Reviewer 2



This paper considered the problem of bilinear inversion of sparse vectors. Under the (somewhat unusual) assumption that the signs of the sparse vectors are also available, it was shown that exact recovery can be achieved via a rather natural convex program. The sample complexity for Gaussian random subspaces was derived based on the Rademacher complexity. I personally feel that the availability of the sign information is somewhat contrived, and I'm not familiar enough with the applications mentioned in the paper. Otherwise, the technical contributions appear to be rather substantial to me.

Reviewer 3



UPDATE AFTER READING THE AUTHORS' RESPONSE: I'm pretty happy with the authors' response -- they provide some reasons why the proof technique is novel, and why the image processing example is noteworthy. Unfortunately, the author response is also necessarily brief, and to be really confident, I'd have to see the revised paper. So, I'm not increasing my score. But subjectively I feel better about the paper now. Summary of the paper: This paper addresses the problem of recovering a pair of sparse vectors from a bilinear measurement. The authors propose a convex programming algorithm for this task. In the case where the signals are sparse with respect to random dictionaries, they prove that their algorithm succeeds with a near-optimal number of measurements. They develop a fast ADMM-type solver for their convex program, and apply it to an image processing problem (removing image distortions). Overall, I think this paper is interesting, and technically impressive. However, note that this paper is building on a line of recent work on bilinear measurements and phase retrieval, and it appears to be more of an incremental improvement. (For instance, it appears that the convex program is a variant of the recently-proposed BranchHull program, augmented with L1-minimization to recover sparse signals.) Some specific strengths and weaknesses of the paper: The recovery guarantee uses some very nice probabilistic techniques, such as Mendelson's small-ball method. However, it only addresses the case of random dictionaries, which is mainly of theoretical rather than practical interest. On the practical side, the paper makes a nice contribution by developing robust versions of the convex program that can handle noisy data, developing a fast ADMM solver, and applying it to an image processing task. One weakness is that there is not much explanation of that image processing task (for instance, what class of image distortions does one want to remove in the real world? and how does the authors' method compare to other methods for doing the same task?). Suggestions for improvements to the presentation: In Theorem 1, can one say more about how the constant C_t depends on t? Section 4 (the proof outline) needs some polishing. In particular, the \oplus notation is not explained when it appears in Lemma 1; it is only explained much later, in a footnote on page 8. The sentence beginning on line 217 is confusing; a better explanation is on line 222. In Lemma 2, the inequality involving L has an extra \tau, compared to the inequality that appears in the proof on page 13 of the Supplemental Material. Also, in line 258, when discussing the bounds on the Rademacher complexity C(D) and the tail probability p_\tau(D), it would be helpful to state the bounds in the main paper, even if the proofs are deferred to the Supplemental Material. Finally, there are several typos and minor grammatical errors throughout Section 4.